# MIXTURE OF NEURON EXPERTS

## ABSTRACT

In this work, We first explore whether the parameters activated by the MoE layer remain highly sparse at inference. We perform a sparsification study on several representative MoE models. For each expert, we rank parameters by the magnitude of their activations from the gate projection and progressively prune the activated subset. Pruning up to $60\%$ of parameters within that subset causes only negligible task-performance degradation; substantial drops occur only after more than $90\%$ are removed. We further decompose experts into neuron granular MoE and visualize their activation values, finding that most neuron activations are near zero. This observation motivates us to select only high-activation neuron experts during pretraining. Based on this insight, we propose *Mixture of Neuron Experts* (MoNE). MoNE achieve neuron granular expert selection by only applying a simple top-$k$ selection within each expert, incurs negligible latency, and requires no additional routing parameters or inter-expert communication. Extensive experiments demonstrate that MoNE matches traditional MoE performance while activating only $50\%$ of the MoE-layer parameters, and it consistently outperforms traditional MoE when compared at equal numbers of activated parameters. These results suggest that MoNE is a practical approach to improving parameter utilization and inference efficiency in MoE-like models.

## 1 INTRODUCTION

Large language models (LLMs) (Dai et al., 2024; Bai et al., 2023; Agarwal et al., 2025; Team et al., 2025) based on Mixture-of-Experts approaches have attracted growing interest in both academic research and industry. The fundamental concept of Mixture-of-Experts (MoE) in large language models entails partitioning a large feed-forward network (FFN) into several smaller subnetworks referred to as experts, where only a subset of expert parameters are activated depending on the input. Unlike dense models that activate all parameters uniformly, MoE models achieve greater computational efficiency through sparse activation patterns.

One key motivation for MoE is the long-observed activation sparsity (Frankle & Carbin, 2018; Fedus et al., 2022a; Frantar & Alistarh, 2023; Frankle et al., 2019) in dense networks: for an given input, only a small fraction of parameters are effectively used. Mixture-of-Experts architectures exploit this property via conditional computation, activating a sparse subset of specialists so as to substantially increase model capacity while maintaining computational efficiency. This further motivates us to propose the following question:

*Are the parameters activated by the MoE layer still highly sparse at inference?*

To answer the question, we performed a sparsification study on a set of representative MoE models. For each expert, we ranked the parameters according to the magnitude of their activations values, which calculated by the gate projection. Then we progressively pruned the weights from the activated subset according to their rank. The results are presented in Figure 1. Notably, across three evaluated models, removing up to 60% of the parameters in this subset led to only negligible declines in task performance, with significant degradation occurring only after more than 90% were pruned. These results suggest that the parameter subset selected by the MoE gating mechanism still highly sparsity at inference.

To further explore the sparsity of MoE, we decompose expert into neuron granular MoE. Then we visualize the activation value for the neuron experts. As shown in Figure 2, most of the activation

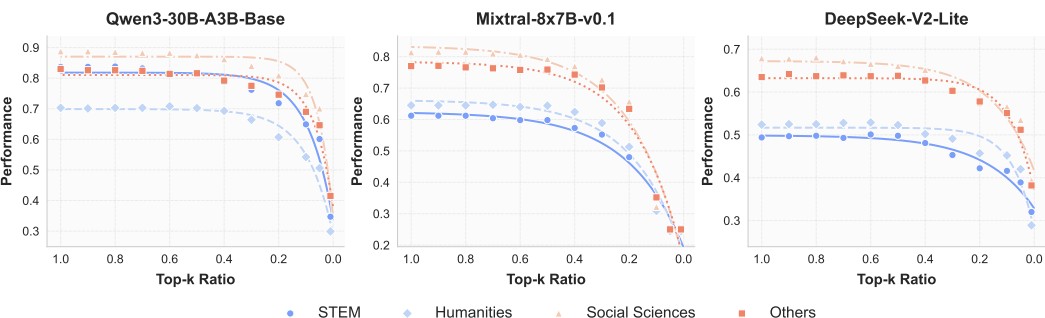

Figure 1: The performance of mainstream MoE models when only use the neuron experts with higher activation weight without extra training. Top-K Ratio refers to the ratio of selected neuron experts.

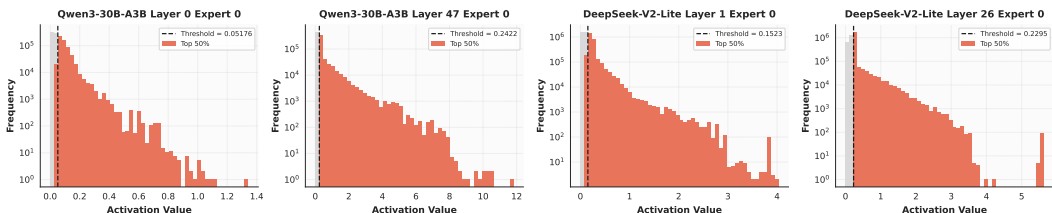

Figure 2: The activation value for the neuron experts, and the top 50% of these values were highlighted.

values are small, which further demonstrate the sparsity of the MoE layer. This results motivate us to only use the neuron experts with high activation weights for pretraining. Also, recently studies have shown the importance of expert granularity (Krajewski et al., 2024; Lepikhin et al., 2020; Du et al., 2022): Deepseek V3 Liu et al. (2024) applies 256 experts, Kimi K2 Team et al. (2025) applies 384 experts, and Qwen3-NextTeam (2025) applies 512 experts. However, overly fine-grained expert partitioning requires substantially larger routing networks and incurs significant cross-device communication latency (Lepikhin et al., 2020; Fedus et al., 2022b). Therefore, we propose Mixture of Neuron Experts (MoNE). By applying a simple top-k selection within each expert, we achieve granular selection for MoE without introducing additional router parameters or inter-expert communication. We evaluate MoNE under multiple settings and find that it consistently outperforms traditional MoE.

Our contributions are summarized as follows:

- We emprically show that traditionally trained Mixture-of-Experts models exhibit high activation sparsity at inference: a small subset of parameters with large activation values retains most of the model's capability, and most of the activation values are small in the MoE layer.
- We introduce *Mixture of Neuron Experts* (MoNE), which decompose the expert into neuron granular MoE, and achieve neuron granular expert selection via a simple top-k within-expert operation that incurs negligible latency overhead and requires no additional routing parameters.
- Extensive experiments demonstrate that MoNE matches the performance of traditional MoE while using only $50\%$ of the parameters in MoE layer. With the same total number of activated parameters, MoNE consistently outperforms traditional MoE.

## 2 RELATED WORK

### 2.1 LARGE LANGUAGE MODELS

Large language models (LLM) (Touvron et al., 2023a; Bai et al., 2023; Brown et al., 2020; Achiam et al., 2023; Liu et al., 2024; Devlin et al., 2019; Raffel et al., 2020) have shown remarkable abil-

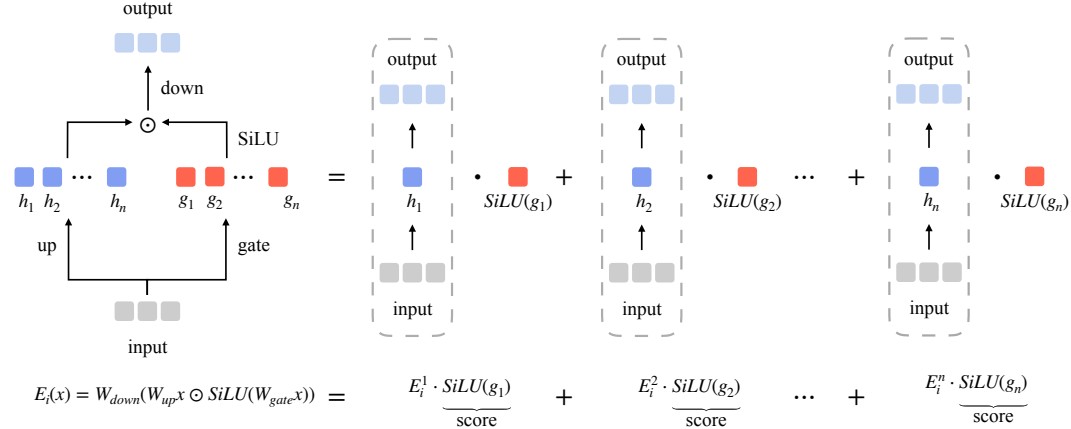

$$E_i(x) = W_{down}(W_{up}x \odot SiLU(W_{gate}x)) \quad = \quad \underbrace{E_i^1 \cdot SiLU(g_1)}_{score} \quad + \quad \underbrace{E_i^2 \cdot SiLU(g_2)}_{score} \quad \cdots \quad + \quad \underbrace{E_i^n \cdot SiLU(g_n)}_{score}$$

Figure 3: Expert in traditional MoE can be decomposed as the weighted sum of neuron granular FFN, which can be realized as a neuron granular MoE.

ities across different tasks, representing important progress toward artificial general intelligence. This success is largely driven by the growth of training data and the expansion of model parameter counts(Wei et al., 2022; Kaplan et al., 2020). And many recent works successfully scaling LLM to billions of parameters (Dai et al., 2024; Liu et al., 2024; Team et al., 2025; Agarwal et al., 2025; Zhang et al., 2022; Scao et al., 2022). However, As model scale increases, the demand for computational resources rises sharply. Consequently, improving the efficiency of both training and inference has become a central research focus to enable further scaling of large language models.

## 2.2 MIXTURE OF EXPERTS

The Mixture of Experts (MoE) (Cai et al., 2025; Masoudnia & Ebrahimpour, 2014; Jiang et al., 2024) architecture was introduced to enhance the capacity of deep neural networks while maintaining computational efficiency. Shazeer et al. (2017) proposed integrating an MoE layer between LSTM layers, demonstrating strong performance in language modeling and machine translation tasks. This approach was later adapted into the transformer framework by replacing the standard feed-forward layers with MoE modules. The Switch Transformer (Fedus et al., 2022b) streamlines the expert selection process by assigning each token to only the top-ranked expert, enabling more efficient model scaling. Gshard (Lepikhin et al., 2020) refined the routing mechanism by employing a Top-2 expert strategy, leading to substantial improvements in multilingual translation across 100 languages. More recently, DeepseekMoE (Dai et al., 2024) and have introduced fine-grained partitioning of experts within the MoE structure. Grove-MoE (Wu et al., 2025) incorporating experts of varying sizes. PEER (He, 2024) scales the number of experts up to one million. Kimi-K2 (Team et al., 2025) scales the model to 1,000B parameters and employs 384 experts, while Qwen3-Next (Team, 2025) further increases the expert count to 512. Improving expert granularity (Tian et al., 2025; Krajewski et al., 2024) and increasing the utilization of activated parameters (Li et al., 2025) are central goals in contemporary MoE research. However, overly fine-grained expert partitioning requires substantially larger routing networks and incurs significant cross-device communication overhead and latency (Lepikhin et al., 2020; Fedus et al., 2022b). PKM (Lample et al., 2019) achieves fast retrieval in a key-value memory layer using a structured memory architecture and further scaled by (Huang et al., 2024). In this work, we analyze the sparsity of activated parameters in traditional MoE architectures and construct a neuron granularity MoE model. This neuron granularity conversion substantially improves the utilization of activated parameters while avoiding the need for a large router and communication latency associated with overly fine expert partitioning in traditional MoE.

## 3 METHOD

### 3.1 PRELIMINARY ON MoE

The Mixture-of-Experts (MoE) layer extends a standard Transformer by replacing a dense feed-forward network (FFN) with a collection of expert FFNs and a routing mechanism. For each input token, the router conditionally dispatches only a sparse subset of experts, so that computation is performed by a small number of specialists rather than the entire FFN. This sparse execution increases the model's effective capacity while keeping per-token computation and latency approximately constant, enabling parameter-efficient scaling to much larger models. (Lepikhin et al., 2020; Dai et al., 2024; Fedus et al., 2022b).

Formally, let $\mathbf{x} \in \mathbb{R}^{d_{\mathrm{model}}}$ denote an input hidden state. An MoE layer consists of $\mathrm{N}_E$ experts $\{E_i\}_{i=1}^{\mathrm{N}_E}$ together with a router that maps $\mathbf{x}$ to routing logits. The experts typically use the *Gated Linear Unit* (Dauphin et al., 2017) structure, which can be formulated as:

$$\mathbf{E}_i(\mathbf{x}) = \mathbf{W}_{\mathrm{down}}^i(\mathtt{SiLU}(\mathbf{W}_{\mathrm{gate}}^i\mathbf{x}) \odot \mathbf{W}_{\mathrm{up}}^i\mathbf{x}) \tag{1}$$

where $\mathbf{W}_{\mathrm{gate}}^i \in \mathbb{R}^{d_{\mathrm{model}} \times d_{\mathrm{expert}}}$ is the gate projection, $\mathbf{W}_{\mathrm{up}}^i \in \mathbb{R}^{d_{\mathrm{model}} \times d_{\mathrm{expert}}}$ is the up projection, and $\mathbf{W}_{\mathrm{down}}^i \in \mathbb{R}^{d_{\mathrm{expert}} \times d_{\mathrm{model}}}$ is the down projection. A common routing pipeline is to first calcuate the scores of the router:

$$\mathbf{P}(\mathbf{x}) = \mathtt{Act}(\mathtt{topK}(\mathrm{Router}(\mathbf{x}))), \tag{2}$$

where $\mathtt{topK}(\cdot)$ masks out all but the top-K routing logits and $\mathtt{Act}(\cdot)$ is the activation function. Then the MoE output is calculated as follows:

$$\mathrm{MoE}(\mathbf{x}) = \sum_{i=1}^{\mathrm{N}_E} \mathbf{P}(\mathbf{x})_i\,\mathbf{E}_i(\mathbf{x}). \tag{3}$$

To encourage balanced utilization across experts, we adopt the commonly used auxiliary load-balance loss:

$$\mathcal{L}_{\mathrm{aux}} = \alpha_{\mathrm{aux}} \cdot \mathrm{N}_E \cdot \sum_{i=1}^{\mathrm{N}_E} \mathbf{f}_i \cdot \mathbf{P}_i, \quad \text{where} \tag{4}$$

$$\mathbf{f}_i = \frac{1}{\mathrm{T}}\sum_{\mathbf{x}\in\mathcal{B}} \mathbb{1}\{i \in \mathtt{argtopK}(\mathrm{Router}(\mathbf{x}))\}, \quad \mathbf{P}_i = \frac{1}{\mathrm{T}}\sum_{\mathbf{x}\in\mathcal{B}} \frac{\mathtt{Act}(\mathrm{Router}(\mathbf{x}))[i]}{\sum_{t=1}^{\mathrm{N}_E}\mathtt{Act}(\mathrm{Router}(\mathbf{x}))[t]} \tag{5}$$

Here, $\mathtt{argtopK}$ get the index of the top-K routing logits, $\mathbf{f}_i$ is the fraction of tokens in the batch $\mathcal{B}$ that are assigned to expert $i$, and $\mathbf{P}_i$ is the average gating weight that the router assigns to expert $i$ (both estimated over a batch of size T). Minimizing $\mathcal{L}_{\mathrm{aux}}$ therefore penalizes experts that are either under-selected or consistently receive low gating weight, encouraging the router to distribute token assignments and gating weights more evenly across experts. The coefficient $\alpha_{\mathrm{aux}}$ controls the regularization strength, and the multiplicative factor $\mathrm{N}_E$ normalizes the objective with respect to the number of experts.

### 3.2 MIXTURE OF NEURON EXPERTS

To explore the sparsity within the activated experts, we decompose the each experts into neuron granular mixture of experts. The output of an expert is formulated as:

$$\mathbf{E}_i(\mathbf{x}) = \mathbf{W}_{\mathrm{down}}^i(\mathtt{SiLU}(\mathbf{W}_{\mathrm{gate}}^i\mathbf{x}) \odot \mathbf{W}_{\mathrm{up}}^i\mathbf{x}) \tag{6}$$

For clarity and compactness, we denote the outputs of the gate projection and the up projection by $\mathbf{G}$ and $\mathbf{H}$, respectively. Concretely,

$$\mathbf{G} = \mathtt{SiLU}(\mathbf{W}_{\mathrm{gate}}^i\mathbf{x}) \in \mathbb{R}^{d_{\mathrm{expert}}}, \quad \mathbf{H} = \mathbf{W}_{\mathrm{up}}^i\mathbf{x} \in \mathbb{R}^{d_{\mathrm{expert}}}. \tag{7}$$

Then Equation (6) can be reformulated as:

$$\mathbf{E}_i(\mathbf{x}) = \mathbf{W}_{\mathrm{down}}^i(\mathbf{G} \odot \mathbf{H}). \tag{8}$$

---

**Algorithm 1:** Mixture of Neuron Experts (MoNE)

1: **Input:** Layer input $\mathbf{x} \in \mathbb{R}^{d_{\text{model}}}$, number of experts $n$, number of selected experts $\mathrm{K}_E$, number of selected neurons for each expert $\mathrm{K}_N$, activation function of router `Act`,
2: **Output:** Layer output $\mathbf{h} \in \mathbb{R}^{d_{\text{model}}}$
3: ▷ Initialize $\mathbf{h} = \mathbf{0}$
4: $\mathbf{p} = \text{Router}(\mathbf{x}) \in \mathbb{R}^n$          // Calculate the scores for each expert
5: $\mathbf{I}_E = \text{argtopK}(\mathbf{p})$            // Select top-$\mathrm{K}_E$ experts
6: $\hat{\mathbf{p}} = \text{Act}(\mathbf{p}[\mathbf{I}_E])$          // Calculate the activated scores
7: **for** each selected expert $i \in \mathbf{I}_E$ **do**
8:  $\mathbf{G}_i = \text{SiLU}(\mathbf{W}_{\text{gate}}^i \mathbf{x})$      // Calculate the output of down projection
9:  $\mathbf{I}_N = \text{argtopK}(\text{Abs}(\mathbf{G}_i))$     // Select top-$\mathrm{K}_N$ neurons
10:  $\tilde{\mathbf{W}}_{\text{up}}^i = \mathbf{W}_{\text{up}}^i[\mathbf{I}_N, :], \tilde{\mathbf{W}}_{\text{down}}^i = \mathbf{W}_{\text{down}}^i[:, \mathbf{I}_N]$  // Select the weights used for calculation
11:  $\tilde{\mathbf{E}}_i(\mathbf{x}) = \tilde{\mathbf{W}}_{\text{down}}^i(\mathbf{G}_i[\mathbf{I}_N] \odot \tilde{\mathbf{W}}_{\text{up}}^i \mathbf{x})$  // Calculate the output of expert $i$
12:  $\mathbf{h} = \mathbf{h} + \hat{\mathbf{p}}[i] \cdot \tilde{\mathbf{W}}_{\text{up}}^i \mathbf{x}$     // Sum the layer output
13: **end for**
14: **return** $\mathbf{h}$

---

Let $\mathbf{W}_{\text{down}}^i[:, k] \in \mathbb{R}^{d_{\text{model}}}$ denote the $k$-th column of $\mathbf{W}_{\text{down}}^i$ and $\mathbf{W}_{\text{up}}^i[k, :] \in \mathbb{R}^{1 \times d_{\text{model}}}$ the $k$-th row of $\mathbf{W}_{\text{up}}$, then we expand the product as follows:

$$\mathbf{E}_i(\mathbf{x}) = \sum_{k=1}^{d_{\text{expert}}} \mathbf{W}_{\text{down}}^i[:, k] \, (\mathbf{G}[k] \cdot \mathbf{H}[k]) \tag{9}$$

$$= \sum_{k=1}^{d_{\text{expert}}} \mathbf{G}[k] \cdot (\mathbf{W}_{\text{down}}^i[:, k] \, (\mathbf{W}_{\text{up}}^i[k, :]\mathbf{x})) \tag{10}$$

$$= \sum_{k=1}^{d_{\text{expert}}} \mathbf{G}[k] \cdot (\underbrace{(\mathbf{W}_{\text{down}}^i[:, k]\mathbf{W}_{\text{up}}^i[k, :])}_{\mathbf{A}_k} \mathbf{x}). \tag{11}$$

Consequently, we can get the decomposition as follows:

$$\mathbf{E}_i(\mathbf{x}) = \sum_{k=1}^{d_{\text{expert}}} \mathbf{G}[k] \cdot \mathbf{A}_k \mathbf{x}, \quad \text{where} \quad \mathbf{A}_k = \mathbf{W}_{\text{down}}^i[:, k]\mathbf{W}_{\text{up}}^i[k, :] \in \mathbb{R}^{d_{\text{model}} \times d_{\text{model}}} \tag{12}$$

Eq. equation 12 shows that each expert can be decomposed into a set of neuron granular experts $\mathbf{A}_k$ that weighted by the neuron level activations $\mathbf{G}$. (**In the subsequent articles, we refer to traditional experts as experts and to neuron granular experts as neuron experts.**) Motivated by this perspective, we explore the distribution of $\mathbf{G}$ in the mainstream MoE models at inference. As shown in Figure 2, the majority of neuron experts receive negligible gate weights: most values of $\mathbf{G}$ are very small, indicating that a large fraction of neurons inside each expert are inactive during inference. To further quantify the impact of these low-activation neurons, we ablate neuron experts whose gate values fall below a threshold and measure the resulting performance. As shown in Figure 1 we find that retaining approximately the top $30\%$ of neuron experts by gate magnitude is sufficient to preserve the bulk of the original performance, which implies that conventionally trained MoE architectures induce high sparsity in the set of activated parameters at inference.

To address this issue, we propose the *Mixture of Neuron Experts* (MoNE). Algorithm 1 formulates the pipeline of MoNE. Concretely, the router first selects a set of experts as usual; for each selected expert $i$, we first calculate the the neuron gating weights $\mathbf{G}$, then we use the neuron experts associated with high absolute gating values to calcuate the ouput of each experts. MoNE converts the traditional MoE into a neuron granular MoE via a simple single, per-expert sort-and-select operation. In contrast, an explicit neuron level routing design in traditional MoE would require a substantially larger router and incur heavy cross-expert communication overhead (Lepikhin et al., 2020). MoNE's selection incurs no additional routing parameters and only accesses the parameters of the expert itself ; because the selected neuron experts communicate within their host expert, the

Table 1: Comparison between traditonal MoE and MoNE with the same number of activated experts. MoNE shows comparable results while only use half of the activated parameters in the MoE Layer.

| Model | ARC-C | BOOIQ | HELLA | LAMBDA | MNLI | PIQA | RACE | SIQA | WINO | WNLI | AVG. |
|---|---|---|---|---|---|---|---|---|---|---|---|
| Traditional MoE 4E/64E | $30.55_{\pm1.35}$ | $56.94_{\pm0.87}$ | $47.78_{\pm0.50}$ | $32.70_{\pm0.65}$ | $34.39_{\pm0.48}$ | $69.53_{\pm0.78}$ | $30.33_{\pm0.65}$ | $39.87_{\pm1.11}$ | $52.80_{\pm1.19}$ | $40.85_{\pm1.69}$ | 43.57 |
| MoNE w/ Random Selection 4E/64E | $27.39_{\pm1.30}$ | $56.79_{\pm0.87}$ | $37.94_{\pm0.48}$ | $27.56_{\pm0.62}$ | $35.49_{\pm0.48}$ | $65.13_{\pm0.81}$ | $30.24_{\pm0.65}$ | $38.84_{\pm1.10}$ | $49.88_{\pm1.19}$ | $43.66_{\pm1.70}$ | 42.69 |
| MoNE w/ TopK Selection 4E/64E | $31.31_{\pm1.35}$ | $53.64_{\pm0.88}$ | $48.08_{\pm0.50}$ | $34.76_{\pm0.66}$ | $33.96_{\pm0.48}$ | $70.02_{\pm0.78}$ | $31.48_{\pm0.66}$ | $39.25_{\pm1.10}$ | $51.78_{\pm1.19}$ | $47.89_{\pm1.71}$ | **44.21** |

Table 2: Comparison between traditonal MoE and MoNE with the same number of activated parameters. For 920M parameter and 2.8M LLMs, MoNE exhibits better downstream performance than MoE models.

| Model | ARC-C | BOOIQ | HELLA | LAMBDA | MNLI | PIQA | RACE | SIQA | WINO | WNLI | AVG. |
|---|---|---|---|---|---|---|---|---|---|---|---|
| *925M Activated 925M* | | | | | | | | | | | |
| Dense | $33.28_{\pm1.38}$ | $58.62_{\pm0.86}$ | $52.07_{\pm0.50}$ | $37.05_{\pm0.67}$ | $33.49_{\pm0.48}$ | $71.27_{\pm0.77}$ | $30.72_{\pm0.66}$ | $40.99_{\pm1.11}$ | $54.22_{\pm1.19}$ | $52.11_{\pm1.71}$ | 46.38 |
| *925M Activated 290M* | | | | | | | | | | | |
| Traditional MoE 4E/64E | $30.55_{\pm1.35}$ | $56.94_{\pm0.87}$ | $47.78_{\pm0.50}$ | $32.70_{\pm0.65}$ | $34.39_{\pm0.48}$ | $69.53_{\pm0.78}$ | $30.33_{\pm0.65}$ | $39.87_{\pm1.11}$ | $52.80_{\pm1.19}$ | $40.85_{\pm1.69}$ | 43.57 |
| MoNE w/o NG-LBL 8E/64E | $30.97_{\pm1.35}$ | $55.75_{\pm0.87}$ | $48.01_{\pm0.50}$ | $33.34_{\pm0.66}$ | $34.44_{\pm0.48}$ | $70.67_{\pm0.78}$ | $29.86_{\pm0.65}$ | $38.89_{\pm1.10}$ | $53.83_{\pm1.19}$ | $49.30_{\pm1.72}$ | **44.51** |
| MoNE w/ NG-LBL 8E/64E | $30.38_{\pm1.34}$ | $59.45_{\pm0.86}$ | $49.51_{\pm0.50}$ | $33.96_{\pm0.66}$ | $34.79_{\pm0.48}$ | $70.89_{\pm0.77}$ | $30.24_{\pm0.65}$ | $39.76_{\pm1.11}$ | $53.67_{\pm1.19}$ | $52.11_{\pm1.71}$ | **45.48** |
| *925M Activated 310M* | | | | | | | | | | | |
| Traditional MoE 6E/64E | $33.02_{\pm1.37}$ | $54.50_{\pm0.87}$ | $49.20_{\pm0.50}$ | $34.48_{\pm0.66}$ | $35.04_{\pm0.48}$ | $71.33_{\pm0.77}$ | $30.91_{\pm0.66}$ | $40.58_{\pm1.11}$ | $53.43_{\pm1.19}$ | $36.62_{\pm1.65}$ | 43.91 |
| MoNE w/o NG-LBL 12E/64E | $30.97_{\pm1.35}$ | $55.99_{\pm0.87}$ | $50.07_{\pm0.50}$ | $35.73_{\pm0.67}$ | $32.35_{\pm0.47}$ | $69.97_{\pm0.78}$ | $30.81_{\pm0.66}$ | $40.43_{\pm1.11}$ | $51.78_{\pm1.19}$ | $52.11_{\pm1.71}$ | **45.02** |
| MoNE w/ NG-LBL 12E/64E | $32.17_{\pm1.36}$ | $62.11_{\pm0.85}$ | $48.65_{\pm0.50}$ | $34.58_{\pm0.66}$ | $33.89_{\pm0.48}$ | $71.44_{\pm0.77}$ | $31.00_{\pm0.66}$ | $39.20_{\pm1.10}$ | $52.88_{\pm1.19}$ | $50.70_{\pm1.72}$ | **45.75** |
| *925M Activated 330M* | | | | | | | | | | | |
| Traditional MoE 8E/64E | $32.68_{\pm1.37}$ | $56.61_{\pm0.87}$ | $49.70_{\pm0.50}$ | $35.51_{\pm0.67}$ | $33.36_{\pm0.48}$ | $71.93_{\pm0.77}$ | $30.33_{\pm0.65}$ | $40.74_{\pm1.11}$ | $51.22_{\pm1.19}$ | $39.44_{\pm1.68}$ | 44.30 |
| MoNE w/o NG-LBL 16E/64E | $31.31_{\pm1.35}$ | $56.39_{\pm0.87}$ | $50.59_{\pm0.50}$ | $35.82_{\pm0.67}$ | $35.36_{\pm0.48}$ | $71.55_{\pm0.77}$ | $31.29_{\pm0.66}$ | $41.30_{\pm1.11}$ | $52.96_{\pm1.19}$ | $52.11_{\pm1.71}$ | **45.82** |
| MoNE w/ NG-LBL 16E/64E | $30.38_{\pm1.34}$ | $60.31_{\pm0.86}$ | $49.17_{\pm0.50}$ | $34.58_{\pm0.66}$ | $34.82_{\pm0.48}$ | $70.84_{\pm0.77}$ | $30.81_{\pm0.66}$ | $40.94_{\pm1.11}$ | $51.38_{\pm1.19}$ | $50.70_{\pm1.72}$ | **45.36** |
| *2.81B Activated 0.55B* | | | | | | | | | | | |
| Traditional MoE 4E/64E (2.81B) | $38.65_{\pm1.42}$ | $57.89_{\pm0.87}$ | $63.00_{\pm0.48}$ | $44.38_{\pm0.69}$ | $31.61_{\pm0.47}$ | $75.19_{\pm0.74}$ | $34.74_{\pm0.68}$ | $42.37_{\pm1.12}$ | $59.98_{\pm1.17}$ | $47.89_{\pm1.71}$ | 50.18 |
| MoNE w/o NG-LBL 8E/64E (2.81B) | $39.93_{\pm1.43}$ | $61.13_{\pm0.86}$ | $63.87_{\pm0.48}$ | $46.26_{\pm0.69}$ | $38.67_{\pm0.49}$ | $76.12_{\pm0.73}$ | $34.70_{\pm0.68}$ | $42.82_{\pm1.12}$ | $59.19_{\pm1.17}$ | $43.66_{\pm1.70}$ | **50.44** |
| MoNE w/ NG-LBL 8E/64E (2.81B) | $37.54_{\pm1.41}$ | $62.97_{\pm0.85}$ | $63.36_{\pm0.48}$ | $46.13_{\pm0.69}$ | $36.28_{\pm0.49}$ | $76.17_{\pm0.73}$ | $34.83_{\pm0.68}$ | $42.32_{\pm1.12}$ | $59.67_{\pm1.17}$ | $57.75_{\pm1.70}$ | **51.72** |

extra communication latency can be negligible. Empirically, pretraining with 50% of the activation parameters by MoNE already matches the performance of a traditional MoE, which demonstrate MoNE can effectively improve the ultilization of activated parameters.

## 3.3 NEURON GRANULAR LOAD BALANCE LOSS

Since We decompose each expert into neuron level sub-experts, we further introduce the *neuron granular load balance loss* (NG-LBL). NG-LBL is designed to avoid cases where a subset of neurons are rarely activated, thereby further improving parameter utilization. The formulation of NG-LBL of is similar with $\mathcal{L}_{\text{aux}}$, but is applied independently to the neuron experts within each expert.

For expert $i$, the fraction of tokens in the batch $\mathcal{B}$ (with batchsize of T) that assigned to neuron $k$, and the average gating weights that the gate projection assigned to neuron $k$ is calculated as follows:

$$\tilde{\mathbf{f}}_{i,k} = \frac{1}{\text{T}} \sum_{\mathbf{x} \in \mathcal{B}} \mathbb{1}\{k \in \texttt{argtopK}(\texttt{Abs}(\mathbf{G}_i))\}, \quad \tilde{\mathbf{P}}_{i,k} = \frac{1}{\text{T}} \sum_{\mathbf{x} \in \mathcal{B}} \frac{\texttt{Abs}(\mathbf{G}_i[k])}{\sum_{t=1}^{d_{\text{expert}}} \texttt{Abs}(\mathbf{G}_i[t])}. \quad (13)$$

The whole auxiliary load balance loss used in MoNE is to sum the orginal auxiliary load balance loss and each expert's neuron granular load balance loss:

$$\tilde{\mathcal{L}}_{\text{aux}} = \mathcal{L}_{\text{aux}} + \sum_{i=1}^{N_E} \mathcal{L}_{\text{NG-LBL}}^i, \quad \text{where} \quad \mathcal{L}_{\text{NG-LBL}}^i = \alpha_{NG} \cdot d_{\text{model}} \cdot \sum_{k=1}^{d_{\text{model}}} \mathbf{f}_{i,k} \cdot \mathbf{P}_{i,k} \quad (14)$$

where the $\alpha_{\text{NG-LBL}}$ is the coefficient to controls the regularization strength of NG-LBL.

## 4 EXPERIMENT

### 4.1 EXPERIMENTAL SETUP

**Model Architectures.** As shown in Table 6, we implement models with total parameter counts of 925M and 2.81B. The MoE layers for both model scales contained 64 experts. Follow by Dai et al.

Table 3: Architecture exploration on different numbers of selected neurons $K_N$. MoNE exhibits better downstream performance when the selected rate is $1/4$.

| Model | ARC-C | BOOIQ | HELLA | LAMBDA | MNLI | PIQA | RACE | SIQA | WINO | WNLI | AVG. |
|---|---|---|---|---|---|---|---|---|---|---|---|
| *2.81B Activated 0.55B* | | | | | | | | | | | |
| Traditional MoE 4E/64E | $38.65_{\pm1.42}$ | $57.89_{\pm0.87}$ | $63.00_{\pm0.48}$ | $44.38_{\pm0.69}$ | $31.61_{\pm0.47}$ | $75.19_{\pm0.74}$ | $34.74_{\pm0.68}$ | $42.37_{\pm1.12}$ | $59.98_{\pm1.17}$ | $47.89_{\pm1.71}$ | 49.57 |
| $K_N = 1/2 \cdot d_{model}$ | | | | | | | | | | | |
| MoNE w/o NG-LBL 6E/64E | $41.04_{\pm1.44}$ | $57.34_{\pm0.87}$ | $63.50_{\pm0.48}$ | $46.21_{\pm0.69}$ | $36.98_{\pm0.49}$ | $75.68_{\pm0.73}$ | $35.12_{\pm0.68}$ | $43.35_{\pm1.12}$ | $59.76_{\pm1.17}$ | $45.07_{\pm1.71}$ | 50.41 |
| MoNE w/ NG-LBL 6E/64E | $39.59_{\pm1.43}$ | $61.96_{\pm0.85}$ | $63.10_{\pm0.48}$ | $44.89_{\pm0.69}$ | $38.41_{\pm0.49}$ | $75.30_{\pm0.73}$ | $35.22_{\pm0.68}$ | $42.17_{\pm1.12}$ | $59.12_{\pm1.17}$ | $45.07_{\pm1.71}$ | **50.48** |
| $K_N = 1/4 \cdot d_{model}$ | | | | | | | | | | | |
| MoNE w/o NG-LBL 8E/64E | $39.93_{\pm1.43}$ | $61.13_{\pm0.86}$ | $63.87_{\pm0.48}$ | $46.26_{\pm0.69}$ | $38.67_{\pm0.49}$ | $76.12_{\pm0.73}$ | $34.70_{\pm0.68}$ | $42.82_{\pm1.12}$ | $59.19_{\pm1.17}$ | $43.66_{\pm1.70}$ | 50.64 |
| MoNE w/ NG-LBL 8E/64E | $37.54_{\pm1.41}$ | $62.97_{\pm0.85}$ | $63.36_{\pm0.48}$ | $46.13_{\pm0.69}$ | $36.28_{\pm0.49}$ | $76.17_{\pm0.73}$ | $34.83_{\pm0.68}$ | $42.32_{\pm1.12}$ | $59.67_{\pm1.17}$ | $57.75_{\pm1.70}$ | **51.70** |
| $K_N = 1/10 \cdot d_{model}$ | | | | | | | | | | | |
| MoNE w/o NG-LBL 10E/64E | $40.27_{\pm1.43}$ | $60.86_{\pm0.86}$ | $63.89_{\pm0.48}$ | $46.09_{\pm0.69}$ | $31.59_{\pm0.47}$ | $75.84_{\pm0.73}$ | $34.55_{\pm0.68}$ | $43.09_{\pm1.12}$ | $58.01_{\pm1.17}$ | $45.07_{\pm1.71}$ | 49.93 |
| MoNE w/ NG-LBL 10E/64E | $39.85_{\pm1.43}$ | $58.10_{\pm0.87}$ | $61.99_{\pm0.48}$ | $44.21_{\pm0.69}$ | $32.07_{\pm0.47}$ | $75.03_{\pm0.74}$ | $35.12_{\pm0.68}$ | $42.12_{\pm1.12}$ | $59.27_{\pm1.17}$ | $57.75_{\pm1.70}$ | **50.55** |

Table 4: The performance of MoNE when applying different activation functions. MoNE exhibits better downstream performance when applying `SiLU` and `Sigmoid`.

| Model | ARC-C | BOOIQ | HELLA | LAMBDA | MNLI | PIQA | RACE | SIQA | WINO | WNLI | AVG. |
|---|---|---|---|---|---|---|---|---|---|---|---|
| `SiLU` | | | | | | | | | | | |
| Traditional MoE 4E/64E | $30.55_{\pm1.35}$ | $56.94_{\pm0.87}$ | $47.78_{\pm0.50}$ | $32.70_{\pm0.65}$ | $34.39_{\pm0.48}$ | $69.53_{\pm0.78}$ | $30.33_{\pm0.65}$ | $39.87_{\pm1.11}$ | $52.80_{\pm1.19}$ | $40.85_{\pm1.69}$ | 43.38 |
| MoNE w/o NG-LBL 8E/64E | $30.97_{\pm1.35}$ | $55.75_{\pm0.87}$ | $48.01_{\pm0.50}$ | $33.34_{\pm0.66}$ | $34.44_{\pm0.48}$ | $70.67_{\pm0.78}$ | $29.86_{\pm0.65}$ | $38.89_{\pm1.10}$ | $53.83_{\pm1.19}$ | $49.30_{\pm1.72}$ | 44.67 |
| MoNE w/ NG-LBL 8E/64E | $30.38_{\pm1.34}$ | $59.45_{\pm0.86}$ | $49.51_{\pm0.50}$ | $33.96_{\pm0.66}$ | $34.79_{\pm0.48}$ | $70.89_{\pm0.77}$ | $30.24_{\pm0.65}$ | $39.76_{\pm1.11}$ | $53.67_{\pm1.19}$ | $52.11_{\pm1.71}$ | **45.48** |
| `Sigmoid` | | | | | | | | | | | |
| Traditional MoE 4E/64E | $30.38_{\pm1.35}$ | $54.79_{\pm0.87}$ | $45.78_{\pm0.50}$ | $33.28_{\pm0.65}$ | $34.44_{\pm0.48}$ | $69.97_{\pm0.78}$ | $30.24_{\pm0.65}$ | $38.87_{\pm1.11}$ | $53.10_{\pm1.19}$ | $43.66_{\pm1.70}$ | 43.15 |
| MoNE w/o NG-LBL 8E/64E | $31.40_{\pm1.36}$ | $49.45_{\pm0.88}$ | $48.93_{\pm0.50}$ | $34.76_{\pm0.66}$ | $35.00_{\pm0.48}$ | $71.33_{\pm0.77}$ | $32.73_{\pm0.67}$ | $39.51_{\pm1.11}$ | $49.57_{\pm1.19}$ | $50.70_{\pm1.72}$ | 44.34 |
| MoNE w/ NG-LBL 8E/64E (Sigmoid) | $30.72_{\pm1.35}$ | $59.79_{\pm0.86}$ | $47.51_{\pm0.50}$ | $33.75_{\pm0.66}$ | $34.80_{\pm0.48}$ | $70.18_{\pm0.78}$ | $32.82_{\pm0.67}$ | $40.17_{\pm1.11}$ | $51.38_{\pm1.19}$ | $53.52_{\pm1.71}$ | **45.46** |
| `Softmax` | | | | | | | | | | | |
| Traditional MoE 4E/64E | $27.93_{\pm1.31}$ | $48.20_{\pm0.88}$ | $36.67_{\pm0.48}$ | $29.82_{\pm0.64}$ | $33.28_{\pm0.48}$ | $66.51_{\pm0.80}$ | $29.96_{\pm0.65}$ | $37.84_{\pm1.10}$ | $49.28_{\pm1.19}$ | $40.66_{\pm1.69}$ | 40.02 |
| MoNE w/o NG-LBL 8E/64E | $28.24_{\pm1.31}$ | $47.52_{\pm0.88}$ | $38.30_{\pm0.49}$ | $30.08_{\pm0.64}$ | $35.00_{\pm0.48}$ | $66.43_{\pm0.80}$ | $30.33_{\pm0.65}$ | $39.25_{\pm1.14}$ | $51.54_{\pm1.19}$ | $42.25_{\pm1.70}$ | 40.96 |
| MoNE w/ NG-LBL 8E/64E | $28.58_{\pm1.32}$ | $48.53_{\pm0.88}$ | $44.96_{\pm0.50}$ | $32.52_{\pm0.65}$ | $35.16_{\pm0.48}$ | $69.91_{\pm0.78}$ | $30.43_{\pm0.66}$ | $38.89_{\pm1.10}$ | $50.59_{\pm1.19}$ | $46.48_{\pm1.71}$ | **41.81** |

(2024), we use one shared expert. The MoE layers in both model scales comprise 64 experts. For the smaller model, we trained traditional MoE with 4, 6 and 8 experts-per-token, which correspond to activated parameters of 290M, 310M and 330M, respectively; For the larger model, we trained a traditional MoE with 4 experts-per-token, corresponding to 0.55B activated parameters. In the main experimental suite, each traditional MoE configuration was paired with a corresponding MoNE: the $K_N$ is set to $d_{model}/4$ and $K_E$ is set two times of corresponding traditional MoE. so that the number of activated parameters is equal between the traditional MoE and MoNE. Please refer Appendix A.1 for detailed training hyper-parameters.

**Data & Tokenizer.** We trained the 925M parameter model on a 50B-token subset of the NeMaTron-CC dataset (Su et al., 2024) and the 2.81B parameter model on a 100B-token subset of the NeMaTron-CC dataset. All text was tokenized with the LLaMA-3-8B tokenizer (Dubey et al., 2024).

**Hyper-Parameters.** The hyper-parameters are selected based on the common practice for dense language models. We replace all FFN layers with MoE layer in the transformer. Please refer Appendix A.1 for detailed training hyper-parameters.

**Benchmarks.** We use the lm-evaluation-harness (Gao et al., 2024) for evaluation. The benchmarks used include ARC-C (Clark et al., 2018), BoolQ (Clark et al., 2019), HellaSwag (Zellers et al., 2019), LAMBADA (Paperno et al., 2016), MNLI (Williams et al., 2017), PIQA (Bisk et al., 2020), RACE (Lai et al., 2017), SIQA (Sap et al., 2019), WinoGrande (Sakaguchi et al., 2021), WNLI (Levesque et al., 2012). For all these benchmarks, we report the zero-shot accuracy.

## 4.2 MAIN RESULTS

**Prior Experiment** We first compare traditional MoE and MoNE with the same number of experts per token $N_E$. For MoNE, we set the number of used neurons $K_N$ to $d_{model}/4$, which reduces the number of MoNE's activated parameters in the moe layer to approximately half of the traditional MoE's. As an additional baseline, we select a random subset of size $K_N = d_{model}/4$ from the neuron experts instead of the TopK strategy to pretrain the model. Table 1 shows that MoNE attains performance comparable to the traditional MoE while using $50\%$ of the activated parameters in the

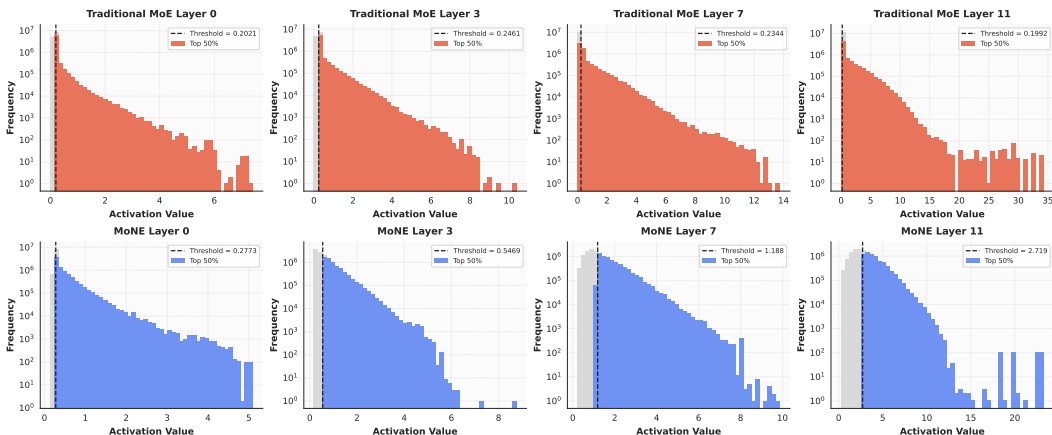

Figure 4: The comparison of the activation value $\mathbf{G}$ for the neuron experts between traditional MoE and MoNE. MoNE effectively increase the activation weight compared with traditional MoE.

MoE layer, whereas the random subset baseline suffers a large drop in performance. These findings indicate that MoNE's selection mechanism effectively improve the utilization of the activated parameters.

**Comparisons to Traditional MoE of Equivalent Activated Parameters** We further compared MoNE with Traditional MoE with equivalent activated parameters. As shown in the Table 2, MoNE consistently improves over the traditional MoE, and the improvement grows as the number of activated parameters increases. Concretely, when activating 290M parameters MoNE yields an improvement of approximately 1% relative to the traditional MoE, and this gain increases to about 2% at 330M activated parameters. Also, with 330M activated parameters, MoNE attains performance that is comparable to a dense model with 925M parameters. For the 3B models MoNE improves upon the traditional MoE by roughly 1.1%. These results indicate that MoNE is a promising approach for training MoE-like models.

### 4.3 FURTHER ANALYSIS

**The Effectiveness of NG-LBL** We investigate the effect of NG-LBL on MoNE in Table 2 and Table 3. Empirically, NG-LBL consistently improves MoNE's performance and increases its parameter efficiency: on tue 1B-parameter model, NG-LBL yields an improvement about $1.0\%$, while on a 3B-parameter model the gain is about $1.4\%$. Figure 5 shows that NG-LBL substantially accelerates the decline of training loss,which demonstrates the effectiveness of NG-LBL. To better understand how NG-LBL helps, we examine load balancing among neuron experts. As shown in Figure 6, neuron experts achieve better load balance with NG-LBL. This indicate that the balancing at the neuron level can effectively increase the ability of experts

**The Analysis of the Activation Weight** Figure 4 compares the activation weight of traditional MoE and MoNE. MoNE effectively increases neuron activation weight: the median value of activation weight in the first layer rises from 0.20 to 0.28 and continues to grow with depth. The median value increases from 0.20 to 2.70 in the final layer. In addition, the activation weight's distribution produced by MoNE is noticeably more uniform, indicating a more balanced utilization of neuron experts. These results demonstrate that MoNE both increases and homogenizes the use of neuron experts, thereby reducing the sparsity of activated parameters and improving the utilization of the activated parameters.

**The Effect of the Neuron Expert Activation Ratio** Figure 1 indicates that the within-expert activation rate has a meaningful effect. Keeping the number of activated parameters fixed, we pretrained models with different $\mathbf{K}_N$; Table 3 shows that all settings improve over the baseline, with the best performance at a ratio of $1/4$. Therefore, we recommend using $\mathbf{K}_N = 1/4 \cdot d_{\mathrm{model}}$. The result aligns with the result in Figure 1: model performance remains stable until approximately $70\%$ of neurons

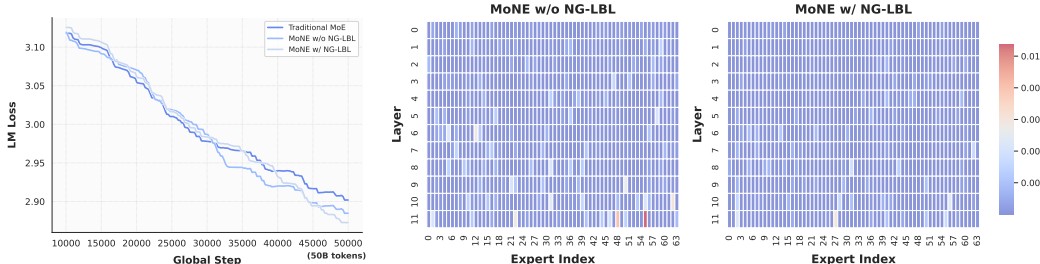

Figure 5: Pre-training loss between traditional MoE and MoNE

Figure 6: The visualization of load balance for different layers and experts, the value visualized is the variance of $\tilde{\mathbf{f}}_{i,k}$ .

are removed, which implies that each activated expert only needs 30% of its neurons to process an input.

**The Effect of Different Activation Function inside the Expert** We further explored three internal activation functions for neuron experts in MoNE—`Sigmoid`, `SiLU`, and `Softmax`. Empirically, `Sigmoid` and `SiLU` produce consistently performance than `Softmax`. While experts contain large amounts of neurons, `Softmax` concentrates probability distribution on a small subset while assigning near-zero weights to most neurons, thereby reducing effective parameter usage. Using NG-LBL mitigates this concentration by encouraging more uniform neuron activation, but `Softmax` still underperforms the baseline in our experiments. Accordingly, we recommend use `Sigmoid` or `SiLU` as the default internal activation for MoNE architectures.

Table 5: Throughput and memory usage comparison among traditional MoE and MoNE. The model with 925M parameters and 290M activation parameters is used for calculation. Auxiliary losses do not impact efficiency.

|  | Traditional MoE | MoNE |
|---|---|---|
| **Configuration** | | |
| Batch size | 8 | 8 |
| Input length | 1024 | 1024 |
| New tokens | 128 | 128 |
| **Throughput & Memory** | | |
| Tokens/sec | 1340.63 | 1338.49 |
| Memory Peak Reserved | 7.8GB | 7.8GB |

**The Efficiency of MoNE** We further investigate the efficiency of MoNE. As shown in Table 5, with the same number of activated parameters, MoNE and traditional MoE require comparable GPU memory and show nearly identical throughput. Crucially, MoNE achieves neuron granular expert selection without enlarging the router or increasing communication latency overhead. Hence, MoNE provides a practical approach to neuron granular expert computation while preserving computation efficiency.

## 5 CONCLUSION

In this work, we demonstrate that the parameters activated by Mixture-of-Experts (MoE) layers is also highly sparse. By decomposing each expert into neuron granular subexperts. we find that many neuron experts receive very small activation weights. The result motivate us to improve the utilization of activated paratmeters by only use the neuron experts with high activation weights. Therefore We propose Mixture of Neuron Experts (MoNE), a simple and practical modification of traditional MoE that operates at neuron granularity: by decomposing experts into neuron granular subexperts and applying a simple sorting operation to the gate-projection outputs prior to expert computation, MoNE converts a traditional MoE into a neuron granular MoE. Furthermore, we propose to apply the neuron granular load-balance loss on the neuron experts to encourage more uniform neuron utilization. MoNE requires no additional model parameters and incurs only a negligible computational overhead relative to traditional MoE. Empirically, MoNE matches baseline performance while activating only half of the parameters in the MoE layer and achieves consistent improvements when compared at equal numbers of activated parameters. Expert granularity is an important focus of current MoE development, while traditional MoE faces problems such as large routers and large

communication delays when expert partitioning is overly fine granularity,. We believe MoNE is a practical step toward more efficient and scalable MoE-like architectures.

## ETHICS STATEMENT

This paper presents work whose goal is to advance the field of large language model. There are many potential consequences of our work, none of which we feel must be specifically highlighted here.

## REPRODUCIBILITY STATEMENT

The details of datasets, model architectures and hyper-parameters are described in Section 4.1 and Appendix A.1.

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

# A  APPENDIX

## A.1  EXPERIMENTAL DATAILS

### A.1.1  MODEL ARCHITECTURES.

We list the model configuration in Table 6. Here we verified the corresponding MoE and MoNE has the same number of activated parameters. Suppose the number of parameters for `gate projection`, `up projection` and `down projection` is N. For a MoE layer with 4 experts activated, the number of activated parameters is $4 \cdot 3 \cdot N$. For a MoNE layer with 6 experts activated and $N_k / d_{\mathrm{model}}$ is $1/2$, the number of activated parameters of an expert can be calculated as $N + 1/2 \cdot 2 \cdot N$, where the parameters of `gate projection` are all activated, and the the parameters of `up projection` and `down projection` only activated $1/2$. Then total activated parameters can be calculated as $6 \cdot (N + 1/2 \cdot 2 \cdot N) = 12N$. Accordingly, for a MoNE layer with 8 experts activated and $N_k / d_{\mathrm{model}}$ is $1/4$, the number of activated parameters is $8 \cdot (N + 1/4 \cdot 2 \cdot N) = 12N$, and for a MoNE layer with 10 experts activated and $N_k / d_{\mathrm{model}}$ is $1/10$, the number of activated parameters is $10 \cdot (N + 1/10 \cdot 2 \cdot N) = 12N$. Therefore the corresponding MoE and MoNE has the same number of activated parameters.

Table 6: **Sizes and architectures of MoNE and traditional MoE models.** "290M/925M" represents an architecture of an approximately 925M parameter, with 290M activated per token during inference.

| Methods | # Layers | # Hidden Size | # Intermediate Size | # Heads | # Head Dim | # The Number of FFN Experts | # The Number of Experts per Token | # $N_k/d_{\mathrm{model}}$ |
|---|---|---|---|---|---|---|---|---|
| Traditional MoE 290M/925M | 12 | 768 | 368 | 16 | 48 | 64 | 4 | - |
| Traditional MoE 310M/925M | 12 | 768 | 368 | 16 | 48 | 64 | 6 | - |
| Traditional MoE 330M/925M | 12 | 768 | 368 | 16 | 48 | 64 | 8 | - |
| Traditional MoE 0.55B/2.81B | 24 | 1024 | 512 | 16 | 96 | 64 | 4 | - |
| MoNE 290M/925M | 12 | 768 | 368 | 16 | 48 | 64 | 8 | 1/4 |
| MoNE 310M/925M | 12 | 768 | 368 | 16 | 48 | 64 | 12 | 1/4 |
| MoNE 330M/925M | 12 | 768 | 368 | 16 | 48 | 64 | 16 | 1/4 |
| MoNE 0.55B/2.81B 6E | 24 | 1024 | 512 | 16 | 96 | 64 | 6 | 1/2 |
| MoNE 0.55B/2.81B 8E | 24 | 1024 | 512 | 16 | 96 | 64 | 8 | 1/4 |
| MoNE 0.55B/2.81B 10E | 24 | 1024 | 512 | 16 | 96 | 64 | 10 | 1/10 |

### A.1.2  HYPER-PARAMETERS.

The hyperparameters are selected based on the common practice for dense transformer language models (Zhang et al., 2024; Geng & Liu, 2023; Touvron et al., 2023b; Xue et al., 2024). The key training hyperparameters used in our experiments are as follows: batch size (tokens) $= 1\,\mathrm{M}$; auxiliary load-balance weight $\alpha_{\mathrm{aux}} = 0.001$; neuron-granular load-balance weight $\alpha_{\mathrm{NG}} = 0.001$; optimizer = FusedAdam (Kingma, 2014); learning rate $= 5e - 4$; router scoring activation function = softmax; weight decay (Loshchilov & Hutter, 2017) $= 0.1$; model maximum sequence length $= 2\mathrm{k}$. These settings were kept fixed across the reported pretraining runs and ablations unless stated otherwise.

### A.1.3  CALCULATE RESOURCES AND ENVIRONMENT

We use deepspeed as the training framework. For the 925M model, We conduct training on a cluster with 4 nodes and 32 A100 GPUs. For the 2.81B model, We conduct training on a cluster with 16 nodes and 128 A100 GPUs.

## A.2 ADDITIONAL EXPERIMENT

### A.2.1 MORE RESULTS ON THE ACTIVATION VALUE FOR THE NEURON EXPERTS.

We visualize additional neuron granular activation values for Qwen3-30B-A3B and DeepSeek-V2-Lite. As shown in Figure 7 and Figure 8, the vast majority of neuron experts receive negligible gate weights: most entries of **G** are close to zero, indicating that a large fraction of neurons within each expert remain effectively inactive during inference.

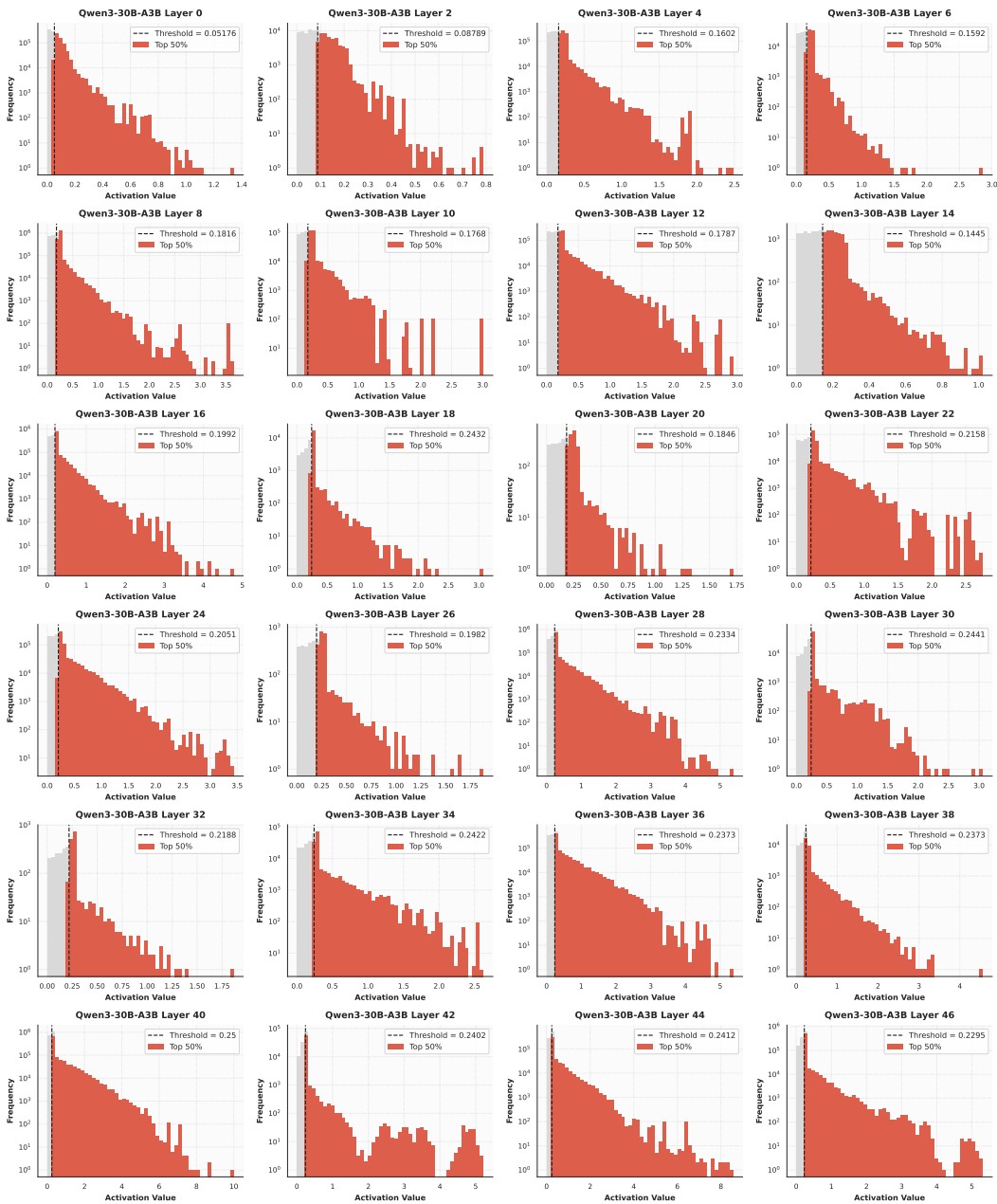

Figure 7: The activation value for the neuron experts on Qwen3-30B-A3

We further compares the activation weight of traditional MoE and MoNE.As shown in Figure 9 and Figure 10 MoNE effectively increases neuron activation weight: the median value of activation weight in the first layer rises from 0.20 to 0.28 and continues to grow with depth. The median

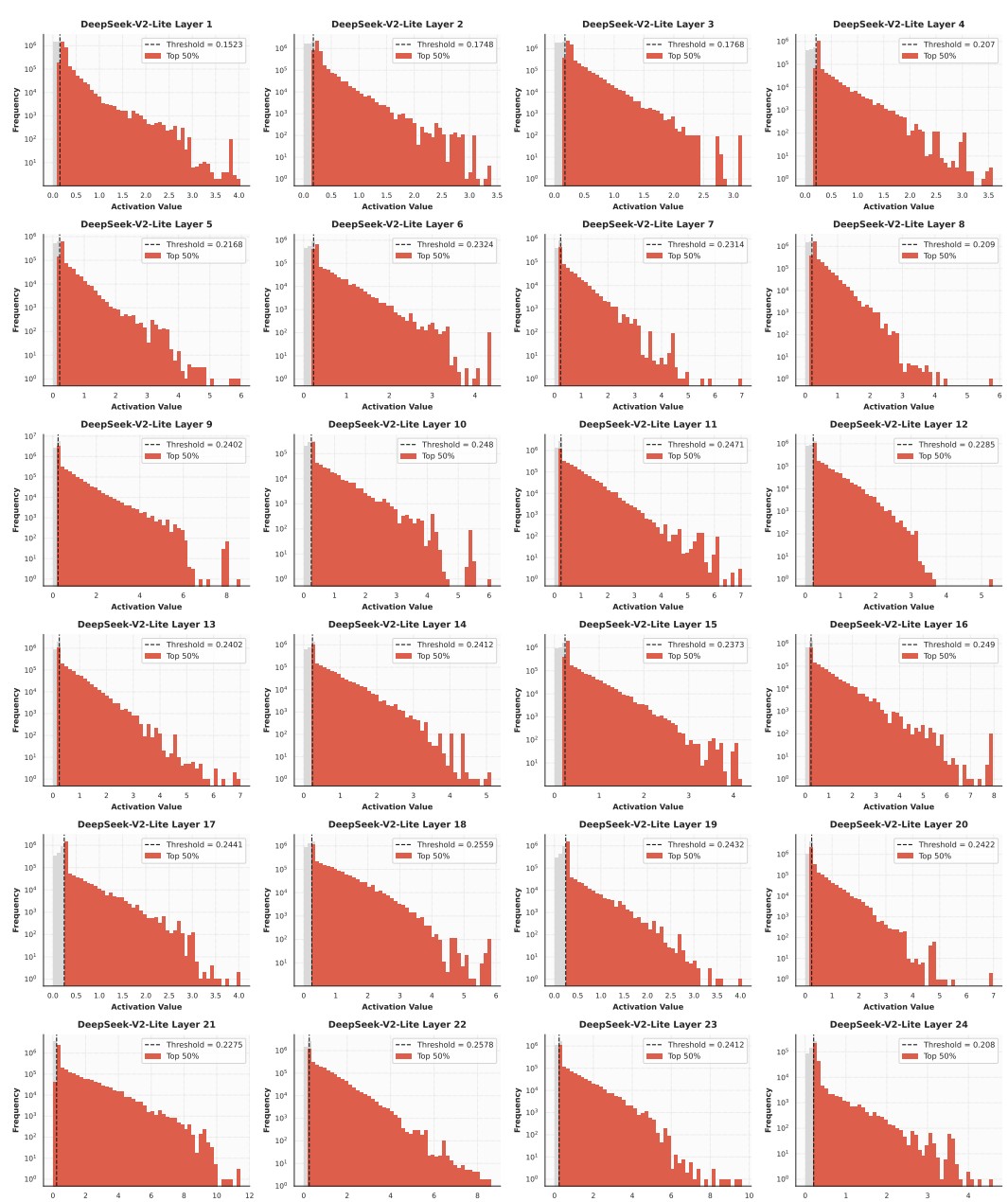

Figure 8: The activation value for the neuron experts on DeepSeek-V2-Lite

value increases from 0.20 to 2.70 in the final layer. In addition, the activation weight's distribution produced by MoNE is noticeably more uniform, indicating a more balanced utilization of neuron experts. These results demonstrate that MoNE both increases and homogenizes the use of neuron experts, thereby reducing the sparsity of activated parameters and improving the utilization of the activated parameters.

### A.2.2 THE COMPARISON OF TRANING LOSS FOR TRADITIONAL MOE AND MONE

As shown in Figure 11, MoNE exhibit more effective expert learning compared with traditional MoE, as evidenced by lower loss values.

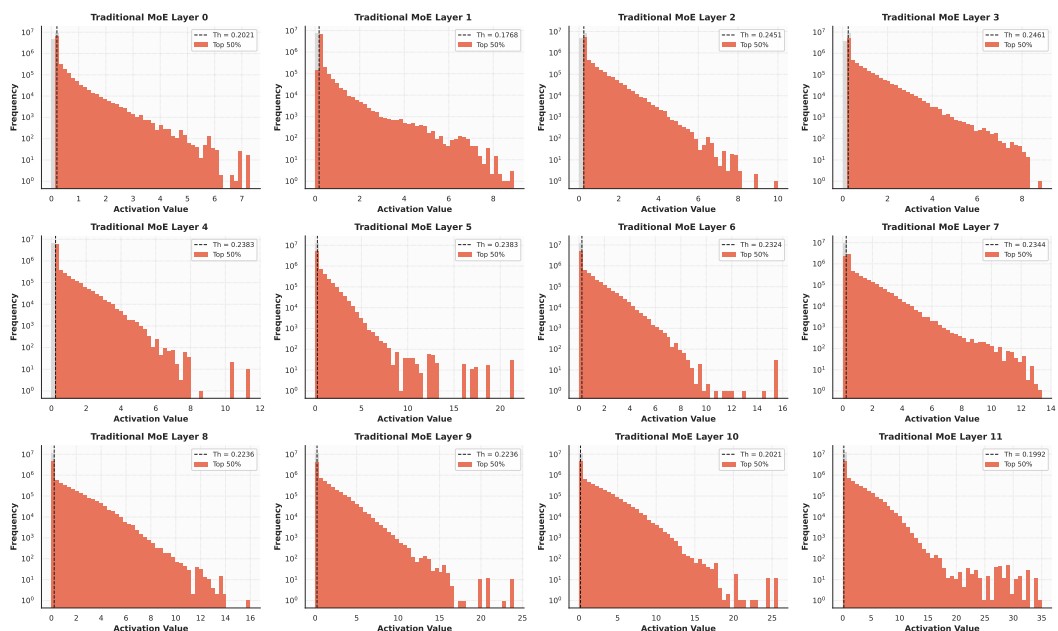

Figure 9: The activation value for the neuron experts on Traditional MoE

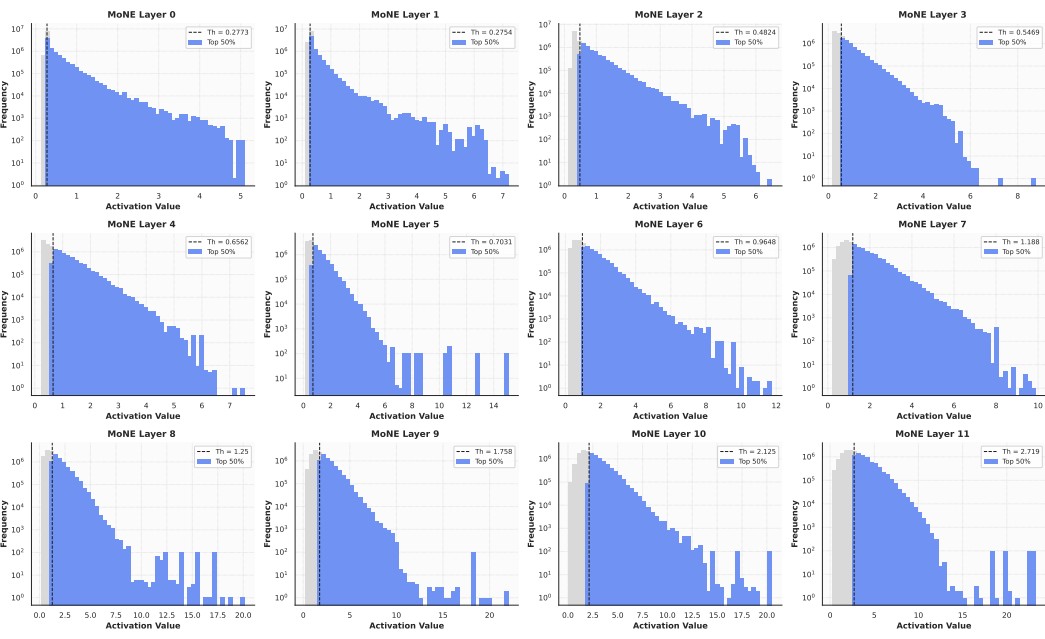

Figure 10: The activation value for the neuron experts on MoNE

### A.2.3 THE EFFECT OF $\mathcal{L}_{\text{AUX}}$ ON MONE

We further investigate the influence of an auxiliary load-balance loss $\mathcal{L}_{\text{aux}}$ on MoNE. Our experimental results show that $\mathcal{L}_{\text{aux}}$ significantly affects MoNE's performance, suggesting that the balacnce across experts is important for MoNE to realize further gains in parameter utilization.

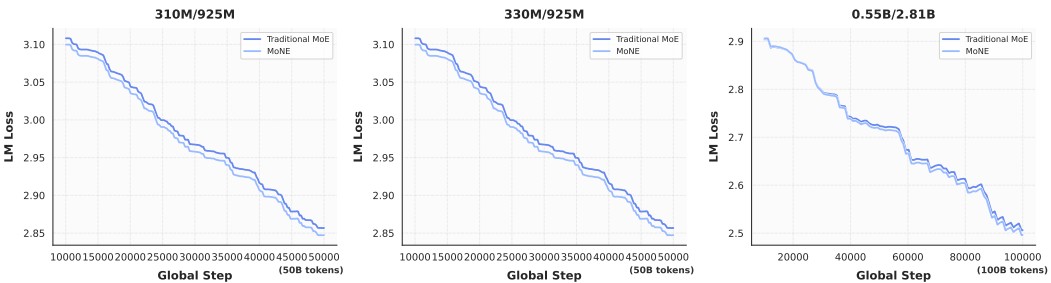

Figure 11: Pre-training loss between tradition MoE and MoNE.

Table 7: The ablation study on auxiliary load-balance loss $\mathcal{L}_{\text{aux}}$

| Model | ARC-C | BOOIQ | HELLA | LAMBDA | MNLI | PIQA | RACE | SIQA | WINO | WNLI | AVG. |
|---|---|---|---|---|---|---|---|---|---|---|---|
| *925M Activated 290M* | | | | | | | | | | | |
| Traditional MoE w/o $\mathcal{L}_{\text{aux}}$ 4E/64E | 28.33 | 46.85 | 45.63 | 33.44 | 34.58 | 68.88 | 30.72 | 38.69 | 52.49 | 50.70 | 43.03 |
| Traditional MoE w/ $\mathcal{L}_{\text{aux}}$ 4E/64E | 30.55 | 56.94 | 47.78 | 32.70 | 34.39 | 69.53 | 30.33 | 39.87 | 52.80 | 40.85 | 43.57 |
| MoNE w/o $\mathcal{L}_{\text{aux}}$ 8E/64E | 28.67 | 52.72 | 44.93 | 32.06 | 34.85 | 69.48 | 30.14 | 39.92 | 53.91 | 42.25 | 42.89 |
| MoNE w/ $\mathcal{L}_{\text{aux}}$ 8E/64E | 30.97 | 55.75 | 48.01 | 33.34 | 34.44 | 70.67 | 29.86 | 38.89 | 53.83 | 49.30 | 44.51 |

## A.3 LLM USAGE

This study utilizes Large Language Models (LLMs) to refine content, adjust formatting, construct tables, and provide writing suggestions for specific chapters.

