# OpenReview forum: "Mixture of Neuron Experts"
_ICLR.cc/2026/Conference — Submitted to ICLR 2026_

### Official Review · Reviewer_rRiW · 2025-10-15

**Soundness:** 2
**Presentation:** 1
**Contribution:** 2
**Rating:** 2
**Confidence:** 4

**Summary:**

This paper proposes a technique to further sparsify the expert parameters in MoE models. Their technique is based on the observation that MoE models which use GLU style experts often have sparse activations in the Gated part of the GLU up-projection which means a subset of the expert weights effectively have null computation. The paper then proposes a technique to only select the top-k activations in the GLU to compute the actual expert output which they call MoNE. They have results on several pretrained MoEs showing this to be the case. Furthermore, they pretrain some models with MoNE and show that this improves performance compared to baseline MoEs.

**Strengths:**

I think the main contribution of the paper is in identifying the effective sparsity in GLU style experts and then being able to decompose the expert computation such that they can run another sparse computation within the expert by only using the outputs of the experts which have high activation values. They were also able to demonstrate this sparse behavior in GLU experts of several pretrained MoE models like Qwen and Mixtral. Finally, they also demonstrated, albeit on a small scale, the benefit of pretraining MoEs with MoNE and showed how it improves upon standard MoE models.

**Weaknesses:**

One of the main weakness of this method is the presentation of the paper. The writing was not effective and I was genuinely confused at several subsections of the paper. The list of errors / confusions I was able to spot are listed below.

Line 18 - achieve neuron granular expert select

Line 47 - magnitude of their activations weights, which calculated by the gate projection.
What are activation weights? I think the authors mean activations values.

Line 256 - experts that associated with high absolute gate

Line 448 - The Effect of Different Activate function inside the Expert

Also, I am fairly confused as to how this is only applicable to MoE models. GLU styled FFNs are common in dense transformers and their technique should be applicable to standard dense models which I believe would serve as a simpler baseline.

The other thing I don't see in the paper is an analysis of the overhead introduced by MoNE. They briefly mention that they do not see any overhead between MoNE and standard MoEs but I don't see how that is happpening. Their is an additional loss computation at each layer, an additional top-k and I am not sure how exactly they have implemented the MoNe computation but since it is token specific and very granular sparsity, I would expect an overhead for that too. However, the details are not provided.

**Questions:**

The authors mention  : "The fundamental concept of Mixture-of-Experts (MoE) in large language
models entails partitioning a large feed-forward network (FFN) into several smaller subnetworks
referred to as experts" .This is only true in the case of fine-grained experts. For most other implementations, it just means we replicate the standard expert N times i.e MoEs are Flop-matched to Dense models during inference / training. Could you please clarify?

Also can you please explain why this technique was studied in the context of MoEs only and not dense transformer models?

---

> ### Author Response · Authors · 2025-11-13
> **Official rebuttal to Reviewer rRiW (Part1/2)**
>
> We sincerely thank you for the in-depth feedback! We highly value each of your comments, and all your concerns are addressed point by point:
>
> ---
> ### Weakness1 — Typos in the paper
>
> We have updated the manuscript to correct the following typos:
>
> - achieve neuron granular expert select → achieve neuron granular expert selection
>
> - magnitude of their activations weights, which calculated by the gate projection. → magnitude of their activation values, which are calculated by the gate projection.
>
> - experts that associated with high absolute gate → neuron experts that are associated with high absolute gate activation values
>
> - The Effect of Different Activate function inside the Expert → The Effect of Different Activation Function inside the Expert
>
> What we want to illustrate here is that after decomposing the expert into a MoE at the neuron level, we only select neuron experts with larger activation weights for computation.  Because this is a MoE decomposing, we've changed the term "activation value" to "activation weight." We've corrected this issue in the text and hope it resolves your concerns.
>
> ------
>
> ### Weakness3 — Implementation of the computation
>
> **Inference-time overhead.**
> First, we want to clarify that the speed we report refers to the inference speed. Compared to the traditional MoE model, we only introduce the additional computation of the topk for an expert, and the auxiliary loss is not calculated during the inference, so the latency during inference is negligible. For an input of 1000 tokens into a single expert (d_model = 768, d_expert = 384, num_selected_neurons = 96), it requires 3 matrix operations:
>
> $
> (1000\times768)\times(768\times384), (1000\times768)\times(768\times96),  (1000\times96)\times(96\times768)
> $
>
> therefore the multiply FLOPs can be calculated as:
>
> $
> \text{FLOPs}_{\text{mul}} =1000\cdot768\cdot384 + 1000\cdot768\cdot96 + 1000\cdot96\cdot768 \approx 3.69\times 10^{8}.
> $
>
> The additional top-K selection in MoNE  costs approximately
>
> $
> \text{FLOPs}_{\text{topk}} \approx 1000 \times 384 \times \log_2(96) \approx 2.53\times 10^{6}.
> $
>
> Therefore the top-K cost is two orders of magnitude smaller than the matrix-multiply cost; in the way the latency overhead introduced by top-K at inference is negligible.
>
>
> **Granularity computation and communication.**
>  Our neuron-level expert selection operates *inside* an expert by selecting neuron subsets of that expert’s weights (see Algorithm 1). Because these selections and computations are local to the expert instance, Also MoNE only introduce negligible additional all-to-all communication compared to conventional MoE. Hence the per-request communication cost remains effectively unchanged.
>
> **Training overhead.**
> In the training stage, since there are the extra loss computation and top-K selection, we observe a modest throughput reduction. On 128 GPUs we measured:
>
> - MoE (925M, 290M activated) — tokens/sec per GPU: 8255
> - MoNE (925M, 290M activated) — tokens/sec per GPU: 7863
>
> Given MoNE achieves neuron-level expert selection and improves model performance at the same activation level, we regard the training throughput reduction as an acceptable trade-off.

---

> ### Author Response · Authors · 2025-11-13
> **Official rebuttal to Reviewer rRiW (Part2/2)**
>
> ### Question1 — The fundamental concept of Mixture-of-Experts (MoE)
>
> We suggest that even when an MoE variant is implemented by replicating a single shared expert N times, it can still be interpreted as a partition or slice of a larger feed-forward network (FFN). This perspective is consistent with prior work — for example, [1] states in the Introduction line 4 that “the core idea of MoE in LLMs involves dividing a large feed-forward network (FFN) into smaller FFNs, known as experts, and activating different experts’ parameters for different inputs.” Practically, mainstream MoE designs (e.g., Switch Transformer, Qwen3-MoE, DeepSeek V3, GPT-OSS) activate only a subset of an expert pool at inference time; consequently, the FLOPs of MoE is different from the FLOPs of a fully dense model.
>
> ------
>
> ###  Weakness2 & Question2 — Why MoNE is studied in MoE context rather than dense transformers
>
> We would like to clarify that our method is not suitable for dense FFN architectures. This is because, for a GLU-based layer, our approach requires computing the entire gate projection before selecting the top-activated neurons. Consequently, even when only one neuron expert is activated, the effective sparsity of the layer remains greater than 1/3 (assuming the gate, up, and down projections have same parameter numbers). This inherent overhead explains why traditional MoE architectures cannot easily achieve neuron-level granularity—too many parameters would need to be processed through the router, while a large portion would not contribute to subsequent computations. Existing large-scale MoE models such as Qwen3-30A3B (128 experts with 8 activated) and DeepSeek V3 (256 experts with 8 activated) achieved a sparsity of much less than 1/3, therefore applying our method directly to the full dense FFN would therefore lead to negative efficiency gains. However, when applied within MoE experts, our approach becomes both practical and efficient: each expert contains far fewer parameters than the full dense FFN, making the cost of calculateing all gate projections acceptable in comparison. Thus, our method is inherently more suitable for MoE architectures rather than dense transformer models.
>
> ---
>
> We thank the reviewer again for their constructive feedback and hope that our detailed responses address the concerns. We respectfully reiterate our core contribution: **achieving neuron-level expert selection without introducing additional router parameters and with negligible communication latency**. Given the current development of MoE research, we believe this neuron-level  selection provides a valuable alternative to merely increasing the number of experts. In light of the clarifications and the manuscript updates described above, we kindly request you to reconsider the score of our paper, your reassessment would be very important to us.
>
> ------
>
> **Reference**
>
> [1] *Autonomy-of-Experts Models*, ICML 2025.

---

> ### Author Response · Authors · 2025-11-26
> **Official Comment by Authors**
>
> Dear reviewer rRiW,
>
> We truly appreciate the time and effort you’ve dedicated to reviewing our submission. We’ve replied to your questions and made revisions. Additionally, we’ve incorporated feedback from other reviewers, which may also help clarify any additional questions you might have.
>
> Since the end of discussion phase is approaching, we wanted to follow up and would greatly value any further thoughts or concerns you might have so we can address them appropriately.
>
> Thank you again for your time and commitment to the review process.
>
> Sincerely,
>
> Authors of Paper 1523

---

### Official Review · Reviewer_n4j5 · 2025-10-28

**Soundness:** 3
**Presentation:** 3
**Contribution:** 3
**Rating:** 8
**Confidence:** 5

**Summary:**

This paper investigates sparsity in Mixture-of-Experts (MoE) architectures at inference and introduces Mixture of Neuron Experts (MoNE), a neuron granular variant of MoE. The authors empirically show that even within the activated experts, most neuron activations are near zero, indicating under-utilization of parameters. MoNE addresses this by applying a top-k neuron selection within each expert, activating only high magnitude neurons while maintaining similar accuracy. They further propose a Neuron-Granular Load-Balance Loss (NG-LBL) to encourage uniform neuron utilization. Experiments on models ranging from 0.9 B to 2.8 B parameters demonstrate that MoNE matches or exceeds the performance of traditional MoE while activating roughly 50 % fewer parameters and incurring negligible latency overhead.

**Strengths:**

1. Comprehensive ablations on activated-parameter ratio (KN), activation functions, and load-balance effects
2. Achieves neuron-granular routing without increasing router complexity or latency.
3.  Visual and quantitative confirmation of intra-expert sparsity.
4. Addresses pressing scalability challenges in sparse-activation LLMs such as Mixtral and DeepSeek.

**Weaknesses:**

1.  The paper does not formalize why top-k neuron selection preserves representational capacity.
2. Experiments rely mainly on NeMaTron-CC (50 B / 100 B tokens); additional pretraining corpora would strengthen generalization claims.
3. Downstream tasks are limited to 10 LM-Eval benchmarks; results on reasoning or multimodal tasks could show broader utility.
4. Some results (e.g., NG-LBL vs baseline) overlap; statistical significance is not reported.

**Questions:**

1. How stable is top-k neuron selection during training—does neuron dropout harm convergence?
2. Could NG-LBL introduce over-regularization for layers with naturally sparse activations?
3. How does MoNE perform when fine-tuned on smaller downstream datasets?

---

> ### Author Response · Authors · 2025-11-20
> **Official rebuttal to Reviewer  n4j5**
>
> We sincerely thank you for the in-depth feedback! We highly value each of your comments, and all your concerns are addressed point by point:
>
> ---
>
>
>
> ### Weakness 1 Why Top-k Neurons Preserve Representation Ability
>
> As shown in Figure 2, most neurons within each expert have activation weights close to zero. Therefore, removing these near-zero activations has minimal impact on the expert’s overall output. Consequently, selecting the Top-k neurons can effectively capture the majority of an expert’s representation information. At the MoE layer level, this means that the overall representation remains largely unchanged, allowing the model to keep the most representation ability even when only the Top-k neurons are used.
>
> ---
>
>
>
> ### Weakness 2&3:   Additional Pretraining Corpus and Downstream Evaluation
>
>
> Due to limited training resources, our experiments were conducted with the current model and corpus scale. Consequently, evaluating the full potential of MoNE models on downstream tasks is constrained by this scale. We would like to emphasize that the current training and evaluation setup is sufficient to demonstrate the effectiveness of MoNE. Recent works [1,2] also conduct their largest-scale experiments on models with ~3B total parameters and 1B activated parameters, trained on ~100B tokens. We sincerely appreciate the reviewer’s suggestion and plan to further scale up MoNE in our future work, enabling it to demonstrate strong performance across a wider range of tasks and datasets.
>
> ---
>
> ### Weakness 4: The statistical uncertainty around the benchmark scores
>
> We have added standard deviation for all results in the main tables, the updated statistical analysis shows that the performance gains reported in the paper are stable.
>
> ---
>
> ### Question1 Stability of Top-k Neuron Selection
>
> We suggest that Top-k neuron selection does not negatively affect training stability. As shown in Figure 5 and Appendix Figure 11, different Top-k settings consistently accelerate the loss decrease. We attribute this to the fact that Top-k retains neurons with the largest activation weights, allowing them to receive more effective gradient updates. In standard MoE, inactive neurons (with near-zero activations) receive almost no gradient during updates, limiting their updating speed. Therefore, Top-k neuron selection not only preserves convergence stability but can also potentially accelerate model convergence.
>
> ---
>
> ### Question2  Does NG-LBL Introduce Over-Regularization for layers with naturally sparse activations
>
> We thank the reviewer for this insightful question. First, we suggest that NG-LBL is designed to encourage a more balanced over usage of neurons across a batch, rather than enforcing strict uniformity at single forward pass. For layers that are naturally sparse, NG-LBL does not disrupt this sparsity; instead, it just ensures that each neuron has a reasonable chance to be selected for the whole time. As shown in Figure 6, NG-LBL does not force every neuron in an expert to be used completely uniform. Rather, it gently redistributes neuron usage to prevent extreme under-utilization while maintaining the natural distribution. Therefore,   NG-LBL  will not  introduce over-regularization for layers with naturally sparse activations.
>
> ---
>
>
>
> ### Question 3: Fine-tuning Results of MoNE on Downstream Tasks
>
> We fine-tuned both the Traditional MoE and MoNE models on *math*  and *code* tasks under identical training setups. We use the 2.8BA0.9B model for finetuning. The results are summarized below:
>
> | Method          | GSM8K | HumanEval |
> | --------------- | ----- | --------- |
> | Traditional MoE | 6.41  | 7.92      |
> | MoNE            | 7.46  | 9.15      |
>
> MoNE consistently outperforms the Traditional MoE after fine-tuning, showing absolute gains of **+1.05** on GSM8K and **+1.23** on HumanEval. This supports our claim that MoNE transfers more effectively and maintains superior performance even after downstream fine-tuning.
>
> ---
>
>
>
> [1] Mixture of Lookup Experts, ICML 2025 Oral
>
> [2] Autonomy-of-Experts Models, ICML 2025

---

### Official Review · Reviewer_R3z4 · 2025-10-30

**Soundness:** 3
**Presentation:** 3
**Contribution:** 2
**Rating:** 6
**Confidence:** 4

**Summary:**

This paper studies sparsity in the activations of Mixture-of-Experts language models. The authors find that the activation patterns are very sparse i.e. most activations are near zero magnitude. The Mixture of Neuron Experts architecture is proposed, introducing sparsity in the activations of experts along with the standard sparse selection of experts themselves in MoEs. The MoNE approach selects the top K neurons from the activations and computes the expert output using only those neurons. The authors show the pruning of these activations has small effect on performance. They also develop a neuron granular load balancing loss to improve the uniformity of expert activations. The authors justify the efficacy of their method with pretraining experiments comparing MoNE to traditional MoE; under the same computational cost they find MoNE outperforms MoE in language modeling.

**Strengths:**

The algorithm is clearly outlined and easy to follow. Diagrams such as Figure 3 are very helpful in visualizing the method.

Empirical findings are well presented, and they evidently motivate the method. The MoNE method is simple and adds relatively little overhead to the MoE architecture.

The Neuron Level Load Balancing Loss is clearly shown to improve pretraining from benchmarks in Table 3. Moreover, even without the load balancing loss the MoNE outperforms standard MoE with the same active parameter count.

**Weaknesses:**

In tables 1-4 the authors do not report statistical uncertainty around the benchmark scores. This is essential to claim the differences are statistically significant.

The method would be more convincing if the authors demonstrated the MoNE preserves performance when fine-tuned on downstream tasks.

The authors show throughput and latency numbers in Table 5, but it is unclear which model scale these numbers are reported for.

The method introduces new hyperparameters, which would require additional tuning in practice

**Questions:**

Could you demonstrate how the throughput and memory are affected by scale? Specifically, does the throughput relative to baseline MoE change as you vary model scale?

Could you provide more detail on hyperparameter selection? How much training is required to select for the optimal hyperparameters?

Could you provide some intuition as to why the activation function affects the results so significantly? Also, are these activation functions compared against the baseline MoE?

---

> ### Author Response · Authors · 2025-11-20
> **Official rebuttal to Reviewer R3z4**
>
> We sincerely thank you for the in-depth feedback! We highly value each of your comments, and all your concerns are addressed point by point:
>
> ---
>
> ### Weakness 1: The statistical uncertainty around the benchmark scores
>
> We have added standard deviation for all results in the main tables in our paper, the updated statistical analysis shows that the performance gains reported in the paper are stable
>
> ---
>
> ### Weakness 2: Fine-tuning Results of MoNE on Downstream Tasks
>
> We fine-tuned both the Traditional MoE and MoNE models on *math*  and *code* tasks under identical training setups. We use the 2.8BA0.9B model for finetuning. The results are summarized below:
>
> | Method          | GSM8K | HumanEval |
> | --- | -- | -- |
> | Traditional MoE | 6.41  | 7.92      |
> | MoNE            | 7.46  | 9.15      |
>
> MoNE consistently outperforms the Traditional MoE after fine-tuning, showing absolute gains of **+1.05** on GSM8K and **+1.23** on HumanEval. This supports our claim that MoNE transfers more effectively and maintains superior performance even after downstream fine-tuning.
>
> ---
>
> ### Weakness 3: Clarification on the Setup for Table 5
>
> We conducted the experiments of table 5 using 925M Activate 290M model, and we have added this clarification to the revised manuscript.
>
> ---
>
> ### Question 1:  How Does the throughput and memory affected by scale
>
> We would like to clarify that model scale does not introduce substantial differences for MoNE. From a computational perspective, MoNE adds only a single additional Top-k operation and does not introduce any additional parameters compared to MoE. The FLOPs introduced by this Top-k step are negligible relative to the matrix multiplications inside each expert.
>
> We further conducted experiments on a larger 2.8BA0.9B MoE/MoNE configuration. The results are shown in the table below. As can be seen, the throughput of MoNE relative to the baseline MoE remains relatively unchanged across different model scales, confirming that model scale does not introduce additional overhead specific to MoNE.
>
> | Method          | Tokens/sec | Memory Peak Reserved |
> | :------- | :--------: | :-----------------: |
> | Traditional MoE |   163.37   |       14.9 GB        |
> | MoNE            |   162.04   |       14.9 GB        |
>
> ---
>
> ### Weakness4 & Question 2: The selection on hyperparameters
>
> We would like to suggest that MoNE is not sensitive to hyperparameter choices. Conceptually, MoNE can be understood as an MoE variant that improves parameter utilization, therefore MoNE can reuse the same training hyperparameters of MoE. Accordingly, **we kept all hyperparameters shared between MoE and MoNE identical in every experiment**, demonstrating that MoNE’s applicability to these hyperparameters matches that of MoE.
>
> Additionally, there are two MoNE-specific hyperparameters in our paper: (1) the number of Top-k neuron,   (2) the coefficient of NG-LBL ;
>
> 1. **The number of Top-k neuron,** As shown in Table 3, MoNE outperforms MoE across different Top-k settings, indicating that MoNE consistently improves MoE’s parameter utilization regardless of the chosen Top-k. Moreover, from the results in Figure 1 and Table 3 we observe that, on average, only about 30% of expert parameters contribute most of the computation. This suggests that setting Top-k around 30% is a sensible default: it preserves MoNE’s benefits while reducing the cost of transferring MoNE to other model scales and tasks.
> 2. **NG-LBL coefficient.** Since MoNE decomposes experts into MoE, the NG-LBL term plays a role similar to the traditional load-balancing loss (LBL). Table 6 shows that using the same coefficient as the traditional LBL already yields effective balancing. Therefore, NG-LBL’s coefficient can be chosen to match the standard LBL setting without extensive hyperparameter search.
>
> ---
>
> ### Question 3: The impact of activation function
>
> We suggest that the primary reason of the observed differences between activation functions lies in the number of expert neurons: each expert contains far more neurons than the number of experts in a traditional MoE. In this way, Softmax concentrates probability distribution on a small subset while assigning near-zero weights to most neurons, thereby reducing effective parameter usage. In contrast, elementwise activations such as SiLU and Sigmoid do not perform across-neuron competition or normalization, so they preserve contributions from a much larger portion of an expert’s neurons. Empirically we observe that SiLU and Sigmoid yield near-identical performance, while softmax-based activations perform noticeably worse, consistent with the argument above.  In the revised manuscript, we have added MoE baselines using Sigmoid and Softmax activations. As shown in table 4, different activation functions in MoE exhibit properties consistent with those observed in MoNE. This also further demonstrates that MoNE consistently provides additional performance improvements over MoE across different activation functions.

---

> ### Author Response · Authors · 2025-11-26
> **Official Comment by Authors**
>
> Dear reviewer R3z4
>
> We truly appreciate the time and effort you’ve dedicated to reviewing our submission. We’ve replied to your questions and made revisions. Additionally, we’ve incorporated feedback from other reviewers, which may also help clarify any additional questions you might have.
>
> Since the end of discussion phase is approaching, we wanted to follow up and would greatly value any further thoughts or concerns you might have so we can address them appropriately.
>
> Thank you again for your time and commitment to the review process.
>
> Sincerely,
>
> Authors of Paper 1523

---

### Official Review · Reviewer_REoY · 2025-10-31

**Soundness:** 2
**Presentation:** 3
**Contribution:** 2
**Rating:** 2
**Confidence:** 5

**Summary:**

This paper proposes Mixture of Neuron Experts (MoNE), which improves the model's parameter efficiency by introducing an additional sparse routing at the neuron level within the MoE experts. The motivation for this approach is derived from observing the sparse patterns within the MoE experts, and the paper introduces a corresponding load balance loss for auxiliary training. Experimental results show that MoNE outperforms MoE in terms of performance with the same number of parameters and active parameters.

**Strengths:**

+ The motivation of this paper is clear, and the presentation is easy to follow.

+ The experiments are comprehensive and demonstrate that there is still redundancy in the active parameters within the MoE.

**Weaknesses:**

+ The paper contains factual error in its description of previous methods, which reduces the credibility of the proposed approach. According to both earlier literature [1] and recent work [2], Equation 5's $P_i = \frac{1}{T} \sum_{x \in B}Act(topK(Router(x)))[i]$ should actually be $P_i = \frac{1}{T} \sum_{x \in B}Act(Router(x))[i]$. This error also extends to the description of the proposed method in the paper, which makes it difficult to verify whether the experimental implementation is correct.

+ The paper introduces a new load balance loss, but the analysis of this loss is insufficient. For example, it is necessary to analyze the distribution of token routing when the loss reaches its minimum in order to better illustrate its effect. In fact, since MoNE's neuron experts use point-wise SiLU or sigmoid as activation functions, the sum of $G_i$ for each token within each expert is not equal to 1. This means that the MoE's load balance loss does not achieve the expected effect in this case. As long as $G_i$ is small, $P_i$ will also be small, and the load balance loss will be minimized, regardless of whether the distribution is balanced. (DeepSeek v3 also uses sigmoid as the router's activation function, but it applies additional normalization when calculating the load balance loss.)

+ The concept of neuron experts in this paper was already introduced in earlier work [3] and scaling attempts were made in [4]. The approach presented in this paper can be seen as a nesting of MoE and Product-Key Models (PKMs), but this is not discussed in the paper.


[1] Switch Transformers: Scaling to Trillion Parameter Models with Simple and Eﬃcient Sparsity

[2] DeepSeek-V3 Technical Report

[3] Large Memory Layers with Product Keys

[4] Ultra-Sparse Memory Network

**Questions:**

See above

---

> ### Author Response · Authors · 2025-11-13
> **Official rebuttal to Reviewer REoY**
>
> We sincerely thank you for the in-depth feedback! We highly value each of your comments, and all your concerns are addressed point by point:
>
> ---
> ###  Weakness1&Weakness2: The typos in  the paper
>
> We sincerely apologize for the oversight. Because of our mistake we inadvertently reused the content from Equation (2), which caused Equations (5) and (14) to be displayed incorrectly. We would like to clarify that both the standard LBL and NG-LBL in our paper adopt the same formulation as the load-balance loss in DeepSeek V3, and we have ensured that the final participating values of P is sum to 1. At this point, the distribution of token routing when the NG-LBL reaches its minimum is consistent with that of the traditional LBL. The manuscript has been updated to reflect these corrections. We hope this resolves your concern.
>
> The revised P in formula 5 is as follows:
>
> $\mathbf{P}_{i}=\dfrac{1}{\text{T}} \sum_x
> \frac{\texttt{Act}(\text{Router} (\mathbf{x}))[i]}{\sum^{\text{N}_E}_t\texttt{Act}(\text{Router} (\mathbf{x}))[t]}$
>
> The revised P in formula 14  is as follows:
>
> $\mathbf{P}_{i,k}=\dfrac{1}{\text{T}} \sum_x
> \frac{\texttt{Abs}(\mathbf{G}_i[k])}{\sum_t^d \texttt{Abs}(\mathbf{G}_i[t])}$
> ###
>
> ---
>
> ### Weakness3: The difference between PKM and MoNE
> We suggest that PKM is essentially an accelerated retrieval mechanism, whereas MoNE leverages the inherent properties of the GLU: a single ToPK selection at the GLU enables expert selection at the neuron level. This design aims to improve parameter utilization and to avoid the excessive all-to-all communication that arises when experts are defined at overly fine granularity. Thus, MoNE is fundamentally different from simply nesting PKM inside an MoE. Furthermore, we suggest that PKM is not a method of MoE, but  an improvement on key-value memory layers. Therefore, it does not incorporate the concept of neuron experts. We also emphasize that our contribution is not the introduction of the “neuron expert” concept. Our main contribution is to decompose the expert into neuron-granular MoE and then achieve neuron-level expert selection by a simple top-K operation without adding extra router parameters or noticeable latency. We have added introduction and citations for PKM and Ultra-Sparse Memory Network in the updated related-work section; we hope this could addresses your concern.
>
> ---
> We thank the reviewer again for their constructive feedback and hope that our detailed responses address the concerns. We would like to gently remind you to focus on our core contribution: decomposing expert into neuron-granular MoE and achieving neuron-level expert selection without introducing additional router parameters and with negligible communication latency. Given current trends in MoE research, we believe this offers a valuable alternative to simply scaling the number of experts. In light of this, we kindly ask you to reconsider the score you assigned, your reassessment means a lot to us.

---

> ### Author Response · Authors · 2025-11-26
> **Official Comment by Authors**
>
> Dear reviewer REoY,
>
> We truly appreciate the time and effort you’ve dedicated to reviewing our submission. We’ve replied to your questions and made revisions. Additionally, we’ve incorporated feedback from other reviewers, which may also help clarify any additional questions you might have.
>
> Since the end of discussion phase is approaching, we wanted to follow up and would greatly value any further thoughts or concerns you might have so we can address them appropriately.
>
> Thank you again for your time and commitment to the review process.
>
> Sincerely,
>
> Authors of Paper 1523

---

> ### Comment · Reviewer_REoY · 2025-11-27
>
> The authors’ correction of the formulas in the paper has addressed my main concerns. However, I still believe that the level of innovation over the Ultra-Sparse Memory Network is somewhat limited, especially since the design of MoNE is quite similar to the PKM baseline in the aforementioned work. Therefore, I am temporarily raising my score to 4 and will participate in the subsequent discussion among reviewers.

---

> ### Author Response · Authors · 2025-11-30
> **Further Response to Reviewer REoY**
>
> We sincerely thank you for reconsidering our work and raising your score. Regarding your concern that the design of MoNE shares similarities with the PKM (Product Key Memory) baseline used in Ultra-Sparse Memory (USM) networks, we would like to offer a detailed clarification.
>
> While both approaches involve conditional computation, we believe MoNE is distinct from PKM/USM in terms of **motivation, fundamental mechanism, architecture, and scientific insight.**
>
> ### 1. Differences in Motivation
>
> - **PKM / USM (Memory Capacity):** These works focus on the **Memory Layer**. Their primary objective is **Expanding Memory Capacity**. By partitioning the memory layer and using retrieval mechanisms, **they aim to decouple model size from computation cost to scale up the number of parameters significantly.**
> - **MoNE (Parameter Utilization):** MoNE focuses specifically on the **MoE architecture**. Our objective is **Exploiting Intrinsic Activation Sparsity** to improve parameter utilization **while keeping the number of parameters constant**. We aim to identify and utilize only the highly active "neuron experts" within a standard expert, thereby reducing redundancy without requiring the massive architectural overhaul associated with memory networks.
>
> ### 2. Differences in Fundamental Mechanism
>
> - **PKM / USM (Key-Query Retrieval):** These architectures replace the standard Feed-Forward Network (FFN) with a Key-Value memory structure. This requires introducing **specific trainable query and key vectors**. Routing is performed via a Nearest Neighbor search (query-key dot products) to retrieve values, which effectively acts as a learned addressing scheme.
> - **MoNE (Activation Magnitude Selection):** MoNE does **not** introduce any separate keys, queries, or routing parameters. Instead, we leverage the intrinsic mathematical properties of the GLU-variant FFN. As derived in **Eq. 12**, we decompose the expert into weighted neuron computations.
>   - **The Innovation:** We discovered that the magnitude of the existing Gate activation is a sufficient proxy for the importance of neuron experts. Consequently, MoNE simply prunes inactive neuron experts based on their scalar activation values.
>   - **Distinction:** Unlike PKM, MoNE does not route to neurons based on Key-Query similarity; it dynamically selects neurons based on their own output magnitude.
>
> ### 3. Differences in Architecture
>
> - **USM:** While USM utilizes a PKM-like baseline, it focuses on **hierarchical routing** to distinct memory blocks to handle massive scale.
> - **MoNE:** MoNE focuses on **within-expert sparsity** inside a standard MoE layer. Mathematically and functionally, MoNE is closer to a dynamic, input-dependent version of network pruning applied at inference time, rather than a Memory Network architecture. It is a drop-in enhancement for existing MoE designs rather than a replacement of the FFN with a memory bank.
>
> ### 4. Scientific Insight: Intrinsic Sparsity
>
> The core contribution of MoNE extends beyond architecture to the empirical findings presented in Figure 1-3:
>
> - We demonstrate that even *within* a selected expert of a trained MoE, activation is highly sparse.
> - We show that 60% of the activated parameters can be removed with only marginal performance degradation.
> - We prove that standard GLU-based experts can be decomposed into neuron-granular MoEs without additional training overheads.
>
> **In conclusion, we contend that MoNE represents a distinct paradigm from PKM/USM.**
>
> ------
>
> **Regarding the Page Limit Policy**
>
> We appreciate your kind reminder regarding the paper length. However, we would like to clarify that our submission adheres to the **ICLR 2026 policy**, which states:
>
> > *"During the discussion/rebuttal phase and for the camera ready, the page limit will be increased to **10 pages** to allow for new results/discussions."*
>
> Therefore, extending the revised version of the paper to 10 pages during this phase is permitted to accommodate the additional experiments and clarifications requested by the reviewers.
>
> ---
>
> We hope these explanations clarify the novelty of MoNE and distinguish it from the PKM/USM lines of research. We are happy to provide further details if needed.

---

### Meta-Review · Area_Chair_f7GK · 2026-01-07

**Summary:**

In their paper, the authors introduce Mixture of Neuron Experts (MoNE), a neuron-granular extension of Mixture-of-Experts (MoE) models that performs top-k selection at the neuron level within each expert to improve parameter utilization and inference efficiency. Reviewers generally agreed that the empirical observation of intra-expert sparsity is interesting, and that the proposed method is simple and easy to integrate into existing MoE architectures. The experimental results demonstrate that MoNE can match or slightly outperform standard MoE under comparable activated-parameter budgets. However, several important weaknesses were raised during the review process that were not sufficiently resolved in the rebuttal.

*Limited novelty:*
Multiple reviewers expressed concerns that the core idea of neuron-level sparsification is conceptually close to existing approaches such as Product-Key Memory and Ultra-Sparse Memory Networks. While the authors argue that MoNE differs in motivation and mechanism, the paper does not convincingly establish a fundamentally new conceptual contribution beyond applying top-k pruning to GLU-style experts. As a result, the novelty relative to prior fine-grained conditional computation and memory-based methods remains unclear.

*Modest performance gains:*
Although MoNE consistently improves parameter efficiency, the reported performance gains over strong MoE baselines are generally modest. Reviewers noted that improvements are often within a narrow margin and depend on specific architectural choices (e.g., activation functions and top-k ratios). Moreover, most experiments are conducted on a limited set of pretraining corpora and model scales, making it difficult to assess the robustness and generality of the reported gains.

*Remaining concerns:*
Some reviewers remained unconvinced that the evaluation fully supports the paper’s claims. In particular, questions were raised about the sensitivity to MoNE-specific hyperparameters, the lack of broader downstream evaluations (e.g., reasoning-heavy or multimodal tasks), and the absence of direct comparisons to dense Transformer baselines where similar neuron-level sparsity might apply. While the rebuttal addressed several points with additional experiments, concerns about the overall fairness and breadth of the evaluation persist.

*Recommendation:*
Given the unclear novelty relative to prior work, modest empirical gains, and unresolved concerns about evaluation fairness, I recommend rejecting the paper in its current form. I encourage the authors to more clearly position their contribution with respect to existing methods and broaden the empirical validation in a future revision.

**Reviewer Concerns:**

Please refer to the summary.

**Reviewer Scores:**

Please refer to the summary.

---

### Decision · Program_Chairs · 2026-01-26

Reject